# Online and Offline Reinforcement Learning by Planning with a Learned Model

**Julian Schrittwieser**[*]
DeepMind
swj@google.com

**Thomas Hubert**[*]
DeepMind
tkhubert@google.com

**Amol Mandhane**
DeepMind
mandhane@google.com

**Mohammadamin Barekatain**
DeepMind
barekatain@google.com

**Ioannis Antonoglou**
DeepMind
University College London
ioannisa@google.com

**David Silver**
DeepMind
University College London
davidsilver@google.com

## Abstract

Learning efficiently from small amounts of data has long been the focus of model-based reinforcement learning, both for the online case when interacting with the environment and the offline case when learning from a fixed dataset. However, to date no single unified algorithm has demonstrated state-of-the-art results in both settings. In this work, we describe the *Reanalyse* algorithm which uses model-based policy and value improvement operators to compute new improved training targets on existing data points, allowing efficient learning for data budgets varying by several orders of magnitude. We further show that *Reanalyse* can also be used to learn entirely from demonstrations without any environment interactions, as in the case of offline Reinforcement Learning (offline RL). Combining *Reanalyse* with the *MuZero* algorithm, we introduce *MuZero Unplugged*, a single unified algorithm for any data budget, including offline RL. In contrast to previous work, our algorithm does not require any special adaptations for the off-policy or offline RL settings. *MuZero Unplugged* sets new state-of-the-art results in the RL Unplugged offline RL benchmark as well as in the online RL benchmark of Atari in the standard 200 million frame setting.

## 1 Introduction

Offline reinforcement learning holds the promise of learning useful policies from many existing real-world datasets in a wide range of important problems such as robotics, healthcare or education (Levine et al., 2020). Learning effectively from offline data is crucial for such tasks where interaction with the environment is costly or comes with safety concerns, but a large amount of logged and other offline data is often available.

A wide variety of effective reinforcement learning (RL) algorithms for the online case have been described, achieving impressive results in video games (Mnih et al., 2015), robotic control (Akkaya et al., 2019) and many other problems. However, applying these online RL algorithms to offline data often remains challenging due to off-policy issues, with the best results in offline RL so far obtained by specialised offline algorithms (Kumar et al., 2020; Wang et al., 2020; Agarwal et al., 2020). At the same time, model-based reinforcement learning (RL) has long focused on learning efficiently from little data, even going as far as learning completely within a model of the environment (Hafner et al., 2018) - an approach ideally suited for offline RL.

---

[*]Equal contribution

35th Conference on Neural Information Processing Systems (NeurIPS 2021).

So far, these developments have been relatively independent, with no unified algorithm that could achieve state-of-the art results in both the online and offline settings.

In this paper, we describe the *Reanalyse* algorithm, a simple yet effective technique for policy and value improvement at any data budget, including the fully offline case. A preliminary version of *Reanalyse* was briefly introduced in the context of MuZero (Schrittwieser et al., 2020), but limited to data efficiency improvements in the discrete action case. Here, we delve deeper into the algorithm and push its capabilities much further – ultimately to the point where most or all of the data is reanalysed.

Starting with the possible uses of *Reanalyse*, we show how it can be used for data efficient learning and offline RL, leading to *MuZero Unplugged*. We demonstrate its effectiveness for the online case through results on Atari and for the offline case through results on the RL Unplugged benchmark for Atari and DM Control.

## 2 Related Work

Recent work by Levine et al. (2020) provides a thorough review of offline RL literature and presents an excellent introduction to the subject. Much research has focused on regularising the value or policy learning to counteract off-policy issues and learn only from high quality data. Critic-Regularized Regression (CRR) uses a critic to filter out bad actions and uses only good actions to train the policy (Wang et al., 2020). Random Ensemble Mixture (REM) regularises q-value estimation by using random convex combinations of ensemble members during training, and the ensemble mean during evaluation (Agarwal et al., 2020). Conservative Q-Learning (CQL) learns a conservative Q-function, used to lower bound the value of the current policy (Kumar et al., 2020). Pessimistic Offline Policy Optimization (POPO) also uses a pessimistic value function for policy learning (He & Hou, 2021).

Existing work has also demonstrated the promise of model-based RL for offline learning (Matsushima et al., 2020; Argenson & Dulac-Arnold, 2020), but has often been restricted to tasks with low-dimensional action or state spaces, and has not been applied to visually more complex tasks such as Atari (Bellemare et al., 2013).

Model-Based Offline Reinforcement Learning (MOReL) implements a two-step procedure, first learning a pessimistic MDP from offline data using Gaussian dynamics models, then a policy within this learned MDP (Kidambi et al., 2020). Results are presented for state-based control tasks. Model-based Offline Policy Optimization (MOPO) penalises rewards by the uncertainty of the model dynamics to avoid distributional shift issues (Yu et al., 2020). Offline Reinforcement Learning from Images with Latent Space Models (LOMPO) extends MOPO to image based tasks (Rafailov et al., 2021). Results are reported on newly introduced datasets with image observations, which the authors aim to open-source in the near future. Deep Averagers with Costs MDP (DAC-MDP) (Shrestha et al., 2021) builds non-parametric models from the offline data, solves these tabular MDPs using value iteration, then generalizes back to the original MDP.

Most previous approaches primarily use the learned model for uncertainty estimation and to train a policy; they do not directly use the learned model for planning over action sequences. In contrast, our method focuses on using the learned model directly for policy and value improvement through planning both offline (when learning from data) and online (when interacting with an environment). It requires no regularisation of the value or policy function either in the online or offline case, works well even in very high dimensional state spaces and is equally applicable to both discrete and continuous action spaces.

Reanalyse is also qualitatively different from Dyna (Sutton, 1991) in several important regards: it uses both value and policy rather than value function alone; and it also updates the state representation. In the specific case of MuZero Reanalyse it also performs a tree search rather than a single step lookahead used in Dyna.

A combination of *MuZero Unplugged* with regularisation approaches such as introduced in the previous work discussed above (Kidambi et al., 2020; Yu et al., 2020; Rafailov et al., 2021) is possible; we leave such investigations for future work.

## 3 Reanalyse

*Reanalyse* takes advantage of model-based value and policy improvement operators to generate new value and policy training targets for a given state (Algorithm 1). In this work, we will use *MuZero*'s Monte Carlo Tree Search (MCTS) planning algorithm combined with its learned model of the environment dynamics as the improvement operator.[2]

As the learned model and its predictions are updated and improved throughout training, *Reanalyse* can be repeatedly applied to the same state to generate better and better training targets. The improved training targets in turn are used to improve the model and predictions, leading to a virtuous cycle of improvement.

---

**Algorithm 1 The *Reanalyse* algorithm**. *MuZero Unplugged* instantiates representation, predict, dynamics with the MuZero network architecture; plan with MCTS; loss with the MuZero loss in eqn (1); and optimise with Adam.

---

$\quad$ **for** step $\leftarrow 0...N$ **do**
$\quad\quad t \sim \text{random}(1:T)$
$\quad\quad s_t^0 = \text{representation}(h_{1:t}, \theta)$
$\quad\quad$ **for** i $\leftarrow 0...k$ **do**
$\quad\quad\quad \pi_t^i, \nu_t^i = \text{plan}(\text{representation}(h_{1:t+i}, \theta), \theta)$
$\quad\quad\quad p_t^i, v_t^i = \text{predict}(s_t^i, \theta)$
$\quad\quad\quad r_t^{i+1}, s_t^{i+1} = \text{dynamics}(s_t^i, a_{t+i}, \theta)$
$\quad\quad$ **end for**
$\quad\quad l = \text{loss}(h_{t:t+k}, \{r, p, v, u, \pi, \nu\}_t^{0:k}, \theta)$
$\quad\quad \Delta\theta = \text{optimise}(l, \theta)$
$\quad$ **end for**

---

To run MCTS and compute new targets for a training point, the representation function of *MuZero* maps the history $h_{1:t}$ of observations, actions and rewards up to timestep $t$ into an agent state or embedding $s_t$. The search over possible future action sequences then takes place entirely in this embedding space, by rolling the dynamics forward and applying prediction functions at every step. These predictions output the key quantities required by planning: the policy, value function and reward. The resulting MCTS statistics at the root of the search tree - visit counts for the actions and value estimate averaged over the tree - are then used as new training targets. During reanalysis, no actions $a$ are selected – instead the agent updates its model and prediction parameters based on the data it has already experienced.

Specifically, *MuZero Reanalyse* jointly adjusts its parameters $\theta$ to repeatedly optimise the following loss at every time-step $t$, applied to a model that is unrolled $0...K$ steps into the future,

$$l_t(\theta) = \sum_{k=0}^{K} l^p(\pi_{t+k}, p_t^k) + \sum_{k=0}^{K} l^v(z_{t+k}, v_t^k) + \sum_{k=1}^{K} l^r(u_{t+k}, r_t^k) \tag{1}$$

where $p_t^k$, $v_t^k$, and $r_k^t$ are respectively the policy, value and reward prediction produced by the $k$-step unrolled model. The respective targets for these predictions are drawn from the corresponding time-step $t+k$ of the real trajectory: $\pi_{t+k}, \nu_{t+k}$ are the improved policy and value generated by the search, $z_{t+k} = u_{t+k+1} + ... + \gamma^{n-1} u_{t+k+n} + \gamma^n \nu_{t+k+n}$ is an $n$-step return, and $u_{t+k}$ is the true reward.

The policy and value predictions are then updated towards the new training targets, in the same way they would be for targets computed based on environment interactions - through minimising losses $l^p$, $l^v$ and $l^r$. In other words, *Reanalyse* requires no changes on the part of the learner and can be implemented purely in terms of adapting the actors to generate improved targets based on stored data instead of environment interactions.

Since the actual MCTS procedure used to *Reanalyse* a state is the same as the one used to choose an action when interacting with an environment, it is straightforward to perform a mix of both. We refer to this ratio between targets computed from direct interactions with the environment, and targets

---

[2]Other model-based algorithms can be used as improvement operators as well.

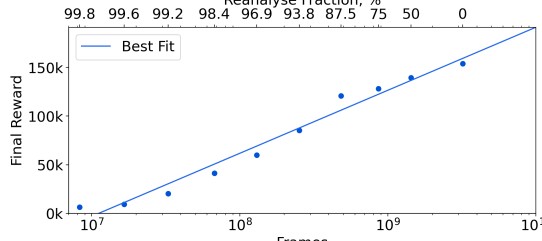

| *Reanalyse* | Median | Mean | # Frames |
|---|---|---|---|
| 50.0% | 1331.7% | 4094.4% | 2000M |
| 95.0% | 1006.4% | 2856.2% | 200M |
| 99.5% | 126.6% | 450.6% | 20M |

Figure 1: ***Reanalyse* scaling in Atari**. By varying the *Reanalyse* fraction alone, *MuZero* can learn efficiently at data budgets differing by orders of magnitude. All other parameters are held constant. Left: Final scores in Ms. Pac-Man for different *Reanalyse* fractions. Note the logarithmic x-axis: Linear improvements in score require exponentially more data, matching scaling laws such as described by (Kaplan et al., 2020) for language models.
Right: Mean & median human normalised scores over 57 Atari games, by *Reanalyse* fraction.

computed by reanalysing existing data points as the *Reanalyse* fraction. A *Reanalyse* fraction of 0% refers to training by only interacting with the environment, no *Reanalyse* of stored data, whereas a fraction of 100% refers to the fully offline case with no environment interaction at all.

Since *Reanalyse* only uses stored data points and the learned model to compute improved targets, it can be employed flexibly for many different purposes:

- **Data Efficiency**. The simplest use of *Reanalyse* is to improve data efficiency by repeatedly computing updated targets on previously collected data throughout training. By scaling the *Reanalyse* fraction as described in Section 4, learning can be optimised for any data budget. For this purpose, the data to be reanalysed is sampled from the $N$ most recent environment interactions; in the limit this includes all interactions throughout training.

- **Offline RL**. When increasing the *Reanalyse* fraction to 100%, learning takes place entirely from stored offline data as described in Section 5, without any interaction with the environment. Offline data may be obtained from a variety of sources, such as other agents, logged data from a heuristic control system or human examples.

- **Demonstrations**. *Reanalyse* can be used to quickly bootstrap learning from demonstrations containing desirable behaviour that might otherwise be hard to discover - collected for instance from humans - while still interacting with the environment, learning from both sources of data at the same time. This is useful to skip past what might otherwise be hard exploration problems while still improving beyond the quality of the initial demonstrations.

- **Exploitation** of good episodes. When using *Reanalyse* to improve data efficiency, *Reanalyse* is applied to the most recently collected data. If instead data is ordered by some other metric, such as episode reward, *Reanalyse* can be used to quickly learn from rare events, such as rewards observed in hard-exploration tasks. This variant is most useful in deterministic environments, as it could otherwise bias the value estimates in stochastic environments.

In this paper, we will focus on the data efficiency and offline RL cases. Remaining cases require no adjustments to the algorithm and only differ in the source of data to be reanalysed. Further combinations of the cases above are also possible, such as a mix of exploitation and data efficiency *Reanalyse* which we leave for future work.

The *Reanalyse* algorithm has some similarities to experience replay (Lin, 1992). Whereas replay performs multiple gradient descent updates for the same data point and target, *Reanalyse* uses model-based improvement operators to generate multiple training targets for the same data point. *Reanalyse* and replay have independent effects and can be combined to further improve data efficiency of learning; in fact we do so for all experiments in this paper.

## 4 Reanalyse for Data Efficiency

By adjusting the ratio between targets computed from interactions with the environment and from stored trajectories (*Reanalyse* fraction), *Reanalyse* can be used to train *MuZero* at any desired data

| Loss | Median | Mean |
|---|---|---|
| a BC | 53.3 % | 48.5 % |
| DQN | 86.2 % | 89.5 % |
| IQN | 100.8 % | 96.1 % |
| BCQ | 107.5 % | 120.0 % |
| REM | 107.9 % | 113.5 % |
| CRR (ours) | 155.6 % | 271.2 % |
| b *MuZero* BC | 54.0 % | 46.9 % |
| *MuZero Unplugged* | **265.3 %** | **595.5 %** |

| Game | QR-DQN | REM | CQL(H) | MZ |
|---|---|---|---|---|
| asterix (1%) | 359.8 | 363.3 | 592.4 | **27220.5** |
| breakout | 6.8 | 4.5 | 61.1 | **251.9** |
| pong | -14.5 | -20.8 | **19.3** | -16.2 |
| qbert | 156.0 | 160.1 | **14012.0** | 6953.2 |
| seaquest | 250.1 | 370.5 | 779.4 | **4964.0** |
| asterix (10%) | 1293.9 | 3912.3 | 156.3 | **40554.0** |
| breakout | 61.8 | 56.9 | 269.3 | **485.8** |
| pong | 12.7 | 9.5 | **18.5** | 15.6 |
| qbert | 9420.5 | 5800.0 | 13855.6 | **16817.9** |
| seaquest | 353.1 | 3643.5 | 3674.1 | **8556.3** |

Table 1: **RL Unplugged Atari benchmark**.
Left: **Overall results**. Mean and median normalised scores over the 46 Atari games of the RL Unplugged benchmark. a) Baseline algorithms. CRR results are for our own reimplementation, other results are from (Gulcehre et al., 2020). b) Results using the *MuZero* network architecture. Behaviour cloning (BC) with the *MuZero* network replicated the baseline BC results from a), confirming correct import of the dataset and evaluation settings. Critic Regularized Regression (CRR) (Wang et al., 2020) significantly improved performance of the policy. *MuZero Unplugged* training with *Reanalyse* loss and MCTS for action selection led to overall best performance.
Right: **Low-data setting**. QR-DQN (Dabney et al., 2018), REM (Agarwal et al., 2020), CQL(H) (Kumar et al., 2020) and *MuZero Unplugged* results when trained on only 1% (top, 2 million frames) or 10% (bottom, 20 million frames) of Atari data. QR-DQN and REM results from (Agarwal et al., 2020). *MuZero Unplugged* performance improves consistently when trained on more data.

budget, as shown in Figure 1. The total amount of computation for each training run (number of updates on the learner and number of searches on the actors) is held constant.

As training progresses, the policy produced by MCTS with the latest network weights will increasingly differ from the policy originally used to generate the trajectories that are being reanalysed. This can bias the state distribution used for training as well as some of the training targets:

- The **policy** prediction $p_t$ for a state $s_t$ is always updated towards the MCTS statistics $\pi_t$ for that same state. In this way, the policy can be learned completely independently from the trajectory; no off-policy issues can arise.
- The **reward** prediction only depends on the state and the action that was taken from this state and is not affected by off-policy issues as such. However, if the state distribution is very biased - in the extreme an action may never be observed - the reward function will be unable to learn the correct reward prediction for these cases, limiting the maximum policy improvement step.
- The situation for the **value** function depends on the choice of training target; when using an n-step TD return such as in Atari ($n = 5$), the target depends on the trajectory and off-policy issues can potentially arise. Whether this is an issue depends on how different the data distribution is from the policy that is being learned. Empirically, we observed that the gain from bootstrapping with the actually observed environment rewards seems to outweigh any harm from being off-policy. We speculate that the bias introduced by early bootstrapping may be larger than the bias introduced by off-policy targets, as also seen in prior work (Vinyals et al., 2019).

## 5  *MuZero Unplugged*: Offline RL with Reanalyse

We obtain *MuZero Unplugged*, an offline version of *MuZero*, by adjusting the *Reanalyse* fraction to 100% - learning without any environment interactions, purely from stored trajectories. In contrast to previous work, we perform no off-policy corrections or adjustments to the value and policy learning: the exact same algorithm applies to both the online and offline case.

We used the RL Unplugged (Gulcehre et al., 2020) benchmark dataset for all offline RL experiments in this paper. To demonstrate the generality of the approach, we report results for both discrete and continuous action spaces as well as state and pixel based data, specifically:

| Loss | supervised | | | CRR | *Reanalyse* |
| Unroll | 0 | 1 | 5 | 5 | 5 |
|---|---|---|---|---|---|
| policy | 60.6 | 61.4 | 54.0 | 155.6 | 203.2 |
| value | 92.2 | 105.0 | 159.2 | 153.0 | 239.9 |
| MCTS | - | 137.3 | 169.7 | 172.5 | 265.3 |

Table 2: **Median score in RL Unplugged Atari: ablations of action selection and training loss**.
Median normalized scores over the 46 Atari games from RL Unplugged.
Rows of the table correspond to different action selection methods: sampling according to the policy probabilities, selecting the action with the highest value or selecting according to MCTS visit counts. Columns correspond to different number of unroll steps of the *MuZero* learned model and different losses. The leftmost three columns use the action from the training data as a supervised policy target, the rightmost two columns use the CRR and the *Reanalyse* loss respectively. For the case of 0 unroll steps, an action-value head is used to predict action values, instead of the state-value predicted by the normal model. All columns use a 5-step TD bootstrap towards a target network as the value target. For all action selection methods, *Reanalyse* loss led to the best performance; for all losses, MCTS action selection also led to the best performance. Overall, the combination of MCTS action selection and *Reanalyse* loss - the *MuZero Unplugged* algorithm - led to the best results.

- **DM Control Suite**, 9 different tasks, number of frames varies by task (Table 3). Continuous action space with 1 to 21 dimensions, state observations.

- **Atari**, 46 games with 200M frames each. Discrete action space, pixel observations, stochasticity through sticky actions (Machado et al., 2017).

*MuZero Unplugged* was highly effective in either setting, outperforming baseline algorithms in Atari (Table 1) as well as the DM Control Suite (Table 3). We performed no tuning of hyperparameters for these experiments, instead using the same hyperparameter values as for the online RL case (Schrittwieser et al., 2020; Hubert et al., 2021).

To add another strong baseline for the Atari benchmark, we also implemented Critic Regularized Regression (CRR), a recent offline RL algorithm (Wang et al., 2020). For the critic value required by CRR we used the value head of *MuZero* model, trained by 5-step TD with respect to a target network, as in previous work (Schrittwieser et al., 2020) and the same as used for *MuZero Unplugged*. Using CRR to train the policy head led to improved results in Atari (Table 1a, CRR), matching results reported for continuous action tasks, but did not reach the same performance as *MuZero Unplugged*.

Performance of *MuZero Unplugged* was robust across the whole range of 46 Atari games in the RL Unplugged benchmark, reaching the same or better performance as the DQN policy used to generate the data in 44 games, and slightly worse performance in only 2 games (Figure 2). Improvements in performance with respect to the training data were considerable, exceeding a 20 times increase in score in several games.

To examine the performance of *MuZero Unplugged* in detail and ascertain the contributions of action selection methods and training losses, we also performed a set of ablations (Tables 2 and 7) based on the Atari dataset. We chose Atari because the large number of diverse levels enables robust performance estimates and its discrete action space allows us to cleanly disentangle the contributions of value and policy predictions as well as planning with MCTS. In contrast, for continuous action spaces such as in the DM Control suite, the contributions of policy and value are entangled, as the value function can only evaluate actions already sampled from the policy.

For our ablations, we considered three possible action selection methods: Sampling actions according to the policy network probabilities, selecting the action with the maximum value, or selecting actions based on the MCTS visit count distribution (rows of Table 2). We also considered different losses and network architectures: the leftmost three columns use variants of the *MuZero* learned model with 0 (no model at all), 1 or 5 steps of model unroll, all trained using the supervised behaviour cloning policy target and a 5-step TD value target based on a target network. The next column used CRR to train the policy. The last column used the the MCTS visit count distribution from the *Reanalyse* loss. These ablations allow us to separately measure the contribution of MCTS at training time (rightmost

column) and evaluation time (bottom row), with the combination of MCTS at evaluation time and *Reanalyse* loss (bottom right cell) corresponding to *MuZero Unplugged*.

As expected, the policy prediction was insensitive to the choice of model depth, but benefited from an improved training target: the CRR loss significantly improved results. Best results were obtained when using the rich MCTS visit count distribution from the *Reanalyse* loss as a training target (top row of Table 2).

When selecting actions according to the value estimate for each action (middle row of Table 2), the depth of the learned model was surprisingly important. The difference between estimating q-values (0-step model) and state-values (1-step model) was small, with both attaining results similar to the IQN baseline (Table 1a) — expected, since all of these results use a distributional value prediction. However, learning a full 5-step model led to a big improvement even though only 1-step value predictions were used for evaluation. We speculate that learning a full 5-step model is beneficial because it regularises the network representation and acts as a useful auxiliary loss.[3]

| Task | # dims | # episodes | Baselines | | | | MuZero | |
| | | | BC | D4PG | BRAC | RABM | BC | *Unplugged* |
|---|---|---|---|---|---|---|---|---|
| cartpole.swingup | 1 | 40 | 386.0 | 856.0 | **869.0** | 798.0 | 143.7 | 343.3 |
| finger.turn_hard | 2 | 500 | 238.0 | **714.0** | 227.0 | 433.0 | 308.8 | 405.0 |
| fish.swim | 5 | 200 | 444.0 | 180.0 | 222.0 | 504.0 | 542.8 | **585.4** |
| manipulator.insert_ball | 5 | 1500 | 385.0 | 154.0 | 55.6 | 409.0 | 412.7 | **557.0** |
| manipulator.insert_peg | 5 | 1500 | 279.0 | 50.4 | 49.5 | 290.0 | 309.9 | **432.7** |
| walker.stand | 6 | 200 | 386.0 | **930.0** | 829.0 | 689.0 | 444.4 | 759.8 |
| walker.walk | 6 | 200 | 380.0 | 549.0 | 786.0 | 651.0 | 496.3 | **901.5** |
| cheetah.run | 6 | 300 | 408.0 | 308.0 | 539.0 | 304.0 | 592.9 | **798.9** |
| humanoid.run | 21 | 3000 | 382.0 | 1.7 | 9.6 | 303.0 | 408.5 | **633.4** |
| mean | | | 365.3 | 415.9 | 398.5 | 486.8 | 406.7 | **601.9** |

Table 3: **Results for DM Control benchmark from RL Unplugged**. Mean final score on 9 DM Control tasks, as well as mean score across all tasks. First three columns indicate task, action dimensonality and dataset size, subsequent four columns reproduce baseline results from (Gulcehre et al., 2020). Final columns show performance of Behaviour Cloning (BC) with the *MuZero* network and results for *MuZero Unplugged*. As the data sets for the DM Control tasks are very small and vary a hundredfold between tasks, to keep the number of model parameters per datapoint constant and prevent memorisation, we scaled the neural network according to $channels = \sqrt{\frac{datapoints}{layers}}$. For an ablation of network size see Table 9.

Keeping the 5-step model but changing the loss for the policy head, we observed that CRR had no effect on the quality of the value prediction for action selection, while the richer MCTS visit count distribution from the *Reanalyse* loss led to another big improvement. Even though the policy head is not used when selecting actions according to the maximum 1-step value, we hypothesise that the auxiliary loss has a strong regularising effect and further improved the internal representation of the model. This matches the results of (Silver et al., 2017) that training a single combined network to estimate both policy and value led to improved value prediction accuracy.

Finally, using MCTS to select actions at evaluation time (bottom row of Table 2) improved results no matter which loss was used at training time, with best results obtained when using MCTS for both training and evaluation - the full *MuZero Unplugged* algorithm.

We also verified that our training setup correctly interpreted the offline data[4] and reproduced the baseline performance when using the same loss: Using the actions played in the training data as

---

[3]These results suggest that an n-step model can also be used as an auxiliary loss to improve the performance of otherwise model-free algorithms. For value-based algorithms without an explicit policy prediction, the distribution used for action selection can be used as the target for the policy loss of the model.

[4]We spent a surprisingly large amount of time tracking down action space mismatches, data discrepancies and compression artefacts. We recommend that any offline RL paper should first reproduce baseline results for the chosen dataset before attempting modifications and improvements to the algorithms.

| Task | CRR | BC | MuZero Unplugged |
|---|---|---|---|
| cartpole.swingup | **664.0** | 501.8 | 594.3 |
| finger.turn_hard | 714.0 | 333.8 | **759.0** |
| fish.swim | 517.0 | 556.8 | **681.6** |
| manipulator.insert_ball | 625.0 | 465.6 | **659.2** |
| manipulator.insert_peg | 387.0 | 325.9 | **556.0** |
| walker.stand | 797.0 | 473.3 | **887.2** |
| walker.walk | 901.0 | 637.9 | **949.5** |
| cheetah.run | 577.0 | 765.3 | **869.9** |
| humanoid.run | 586.0 | 416.5 | **643.1** |
| mean | 640.9 | 497.4 | **733.3** |

Table 4: **Comparison of *MuZero Unplugged* and CRR**. Results for CRR (Wang et al., 2020) were reported by selecting the checkpoint with the highest mean reward from each training run. Since this does not follow the offline policy selection guidelines from RL Unplugged and is therefore not directly comparable to the baseline results, we compared to it separately. The same highest mean reward evaluation scheme as used in CRR was used for *MuZero Unplugged* results in this table as well. All other tables report results at the end of training.

a supervised policy target to train a policy head using cross-entropy loss and sampling from it for evaluation (Table 1, policy BC a and b) reproduced the behaviour cloning (BC) baseline results.

## 6 Offline RL and Continuous Action Spaces

An important motivation for offline RL is the application to real-world systems such as robotics, which often have continuous and high-dimensional action spaces. To investigate the applicability of *MuZero Unplugged* to this setting, we used the DM Control Suite dataset from the RL Unplugged dataset. DM Control is a collection of physics based benchmark tasks (Tassa et al., 2018) with a variety of robotic bodies of different action and state dimensionalities (Table 3).

In order to use planning and *Reanalyse* with continuous action spaces, we used the sample based search extension of *MuZero* introduced by (Hubert et al., 2021). This extension uses a policy head to produce a set of candidate actions to search over, where the MCTS considers only the sampled actions instead of fully enumerating the action space. Finally, the policy is updated towards the search distribution only at the sampled actions.

When applying *Reanalyse* for data efficiency improvements to data generated by the agent itself, no modifications are required to use sample based search and *Reanalyse* together. In offline RL or when reanalysing demonstrations from a source other than the agent itself, the policy that generated the actions making up the dataset is often quite different from the one learned by *MuZero Unplugged*, and unlikely to sample the same actions, at least at the beginning of training. Since in this case the MCTS (and by extension, *Reanalyse*) can only consider actions that have been sampled from the policy, it would be unlikely to learn about the actions contained in the dataset, and thus unable to sample them from the policy in the future. This effect is most pronounced in very high dimensional action spaces.

To prevent this issue, we explicitly included the action from the trajectory being reanalysed in the sample of actions searched over at the root of the MCTS tree. This serves the same purpose as the Dirichlet exploration noise used in standard *MuZero* - encouraging the MCTS to explore actions it would not otherwise consider. For the prior of the injected action we therefore use the same value as for the Dirichlet probability mass, 25%, though the algorithm is not sensitive to the exact value. This step is redundant for discrete action spaces (such as in Atari) where the policy already always produces a prior for all possible actions.

We compared the performance of *MuZero Unplugged* to offline RL algorithms from the literature such as D4PG (Barth-Maron et al., 2018), BRAC (Wu et al., 2019) and RABM (Siegel et al., 2020; Gulcehre et al., 2020) (Table 3), as well as the recent Critic Regularized Regression (CRR) (Wang

et al., 2020) algorithm (Table 4, shown separately as CRR was evaluated by selecting the maximum performance throughout training and results are thus not comparable to the other baselines).

We first measured the performance of Behaviour Cloning (BC) when implemented using the *MuZero* network to ensure we used the offline dataset correctly and that it matches the evaluation environment. Overall performance indeed approximately matches the BC baseline (Table 3).

*MuZero Unplugged* outperformed baseline algorithms both in individual tasks and for the mean return[5] averaged across all tasks. It did best in difficult high-dimensional tasks such as humanoid.run or the manipulator tasks, classified as "hard" by (Wang et al., 2020), compared to "easy" for the other tasks. Performance in the simplest tasks, especially cartpole, was somewhat lower — primarily due to the very small datasets[6] leading to overfitting of the learned model and value function throughout training: in cartpole, performance of the best checkpoint (Figure 4) was much better than performance at the end of training (Figure 3). Additional regularisation techniques such as dropout (Hinton et al., 2012) could be employed to prevent this. We leave this for future work since we are primarily interested in performance on complex tasks that we consider most representative of real-world problems.

# 7 Limitations

*MuZero Unplugged* uses a deterministic model, potentially limiting its performance in stochastic or partially observed environments. The learned model is a single time-step model, which may limit the time horizon of planning. *MuZero Unplugged* also does not employ explicit forms of regularizations; combination with existing methods from the literature may improve its performance on very small datasets.

The MCTS improvement operator in *MuZero Unplugged* requires a suitable value function; in environments where value learning is very difficult this may limit the magnitude of the improvement obtained by Reanalyse.

# 8 Conclusions

In this paper we have investigated the *Reanalyse* algorithm and its applications to both data efficient online RL at any data budget and completely offline RL. We combined *Reanalyse* with *MuZero* to obtain *MuZero Unplugged*, a unified model-based RL algorithm that achieved a new state of the art in both online and offline reinforcement learning. Specifically, *MuZero Unplugged* outperformed prior baselines in the Atari Learning Environment both using a standard online budget of 200 million frames and other data budgets spanning multiple orders of magnitude. Furthermore, *MuZero Unplugged* also outperformed offline baselines in the RL Unplugged benchmark for Atari and continuous control. Unlike previous approaches, *MuZero Unplugged* uses the same algorithm for multiple regimes without any special treatment for off-policy or offline data.

This work represents a further step towards the vision of a single algorithm that can address a wide range of reinforcement learning applications, extending the capabilities of model-based planning algorithms to encompass new dimensions such as online and offline learning, using discrete and continuous action spaces, across pixel and state-based observation spaces, in addition to the wide array of challenging planning tasks addressed by prior work (Silver et al., 2018).

## Acknowledgements

We would like to thank Caglar Gulcehre for providing very detailed feedback and helpful suggestions to improve the paper.

All of the work in this paper was funded by DeepMind.

---

[5]Using the mean is appropriate in DM Control as the return for all tasks is in $[0, 1000]$, with the return for the optimal policy close to 1000. Therefore no outlier can dominate the mean; this is unlike the situation in Atari where scores of wildly varying magnitude require usage of the median.

[6]The amount of information contained in the dataset is the product of the number of episodes and the size of the state and action space.

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
