| Algorithm | Median | Mean | Frames |
|---|---|---|---|
| IMPALA[1] | 191.8% | 957.6% | 200M |
| Rainbow[2] | 231.1% | – | 200M |
| UNREAL[3] [a] | 250.0%[a] | 880%[a] | 250M |
| LASER[4] | 431.0% | – | 200M |
| *MuZero*[5] | 741.7% | 2183.6% | 200M |
| *MuZero* sticky | 692.9% | 2188.4% | 200M |
| *MuZero* Res2 Adam | **1006.4%** | **2856.2%** | 200M |

Table 5: ***MuZero* improvements in Atari**. Mean and median human normalized scores over 57 Atari games, best results are highlighted in **bold**. For per game scores, see Table 10. Top of the table shows results for previously published work, bottom row show results for our experiments. Sticky actions do not significantly hurt performance, and switching to ResNet v2 style pre-activation residual blocks with Layer Normalisation and training with Adam significantly improves performance.
[1] (Espeholt et al., 2018), [2] (Hessel et al., 2018), [3] (Jaderberg et al., 2016), [4] (Schmitt et al., 2019), [5] (Schrittwieser et al., 2020)
[a]Hyper-parameters were tuned per game.

| *Reanalyse* | TD steps | Median | Mean | # Frames |
|---|---|---|---|---|
| 99.5% | 5 | 126.6% | 450.6% | 20M |
| 99.5% | 0 | 115.3% | 385.8% | 20M |

Table 6: **Comparison of value targets in Atari**. Mean and median human normalized scores over 57 Atari games, comparing a 5-step TD update towards a target network compared with direct regression against the search value for a state ("0-step TD"). All other parameters are held constant. Even in the 20M frame setting, where learning is almost entirely off-policy from reanalysed data and a trajectory independent value target might be expected to give better results, bootstrapping along the trajectory (TD 5) was better.

## A  MuZero Implementation

All experiments in this paper are based on a JAX (Bradbury et al., 2018) implementation of *MuZero*, closely following the description in (Schrittwieser et al., 2020). For experiments in environments with continuous actions, we used the extension to *MuZero* proposed in (Hubert et al., 2021). To facilitate a more direct comparison with other algorithms, we used the same Gaussian policy representation as used for data generation and baselines in RL Unplugged (Gulcehre et al., 2020).

The original *MuZero* did not use *sticky actions* (Machado et al., 2017) (a 25% chance that the selected action is ignored and that instead the previous action is repeated) for Atari experiments. To make comparisons against other algorithms easier and match the data from RL Unplugged, our implementation did use sticky actions. As shown in Table 5 (*MuZero* vs *MuZero* sticky), this did not affect performance despite introducing slight stochasticity into the environment.

Additionally, we updated the network architecture of the *MuZero* learned model to use ResNet v2 style pre-activation residual blocks (He et al., 2016) coupled with Layer Normalisation (Ba et al., 2016) and use the Adam optimiser (Kingma & Ba, 2015) with decoupled weight decay (Loshchilov & Hutter, 2017) for training. This significantly improved both mean and median normalized performance, setting a new state of the art for Atari at the 200 million frame budget - see Table 5 for details.

## B  Network Architecture

For all experiments in this work we used a network architecture based on the one introduced by *MuZero* (Schrittwieser et al., 2020), but updated to use use ResNet v2 style pre-activation residual blocks (He et al., 2016) coupled with Layer Normalisation (Ba et al., 2016).

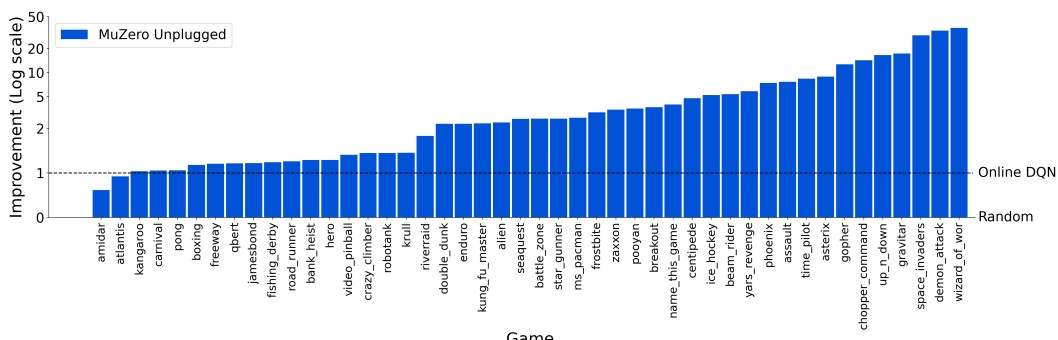

Figure 2: **Atari performance improvement.** Improvement of performance with respect to the online DQN version that was used to generate the training data for the 46 Atari games from RL Unplugged, calculated as $s_{\text{normalized}} = \frac{s_{\text{agent}} - s_{\text{random}}}{s_{\text{dqn}} - s_{\text{random}}}$. 0 is random performance, 1 is the same performance as the training data, and larger than 1 represents an improvement. *MuZero Unplugged* shows very robust performance, reaching the same or higher score in 44 games, with a small decrease in only 2 games.

Both the representation function and the dynamics function were implemented by a ResNet with 10 blocks, each block containing 2 layers. For image inputs each layer was convolutional with a kernel size of 3x3 and 256 planes; for state based inputs each layer was fully-connected with a hidden size of 512.

To implement the network, we used the modules provided by the Haiku neural network library (Hennigan et al., 2020).

## C  Policy Representation

For domains with discrete action spaces, i.e. Atari, we used the same categorical policy representation as *MuZero*, trained by minimising the KL-divergence.

In continuous action spaces, we used the Gaussian policy representation used for the data generation policies in RL Unplugged. We did not observe any benefit from using a Gaussian mixture, so instead in all our experiments we used a single Gaussian with diagonal covariance. We trained the Gaussian policy by maximising the log-likelihood of the training target: for behaviour cloning the log-likelihood of the action from the trajectory; for *Reanalyse* the log-likelihood of each searched action sample, weighted by its normalized visit count.

## D  Value Learning

We follow the approach described in previous work for value learning as well.

In Atari, we followed the *MuZero* training and use Temporal Difference (TD) learning with a TD step size of 5 towards a target network that is updated every 100 training steps by copying the weights of the network being trained (Schrittwieser et al., 2020).

For DM Control, we followed (Hubert et al., 2021) and directly regressed the value prediction for a state against the MCTS value estimate for that state. This allows the value to be learned independently from the trajectory and is helpful to prevent overfitting, useful when only very little training data is available. When more training data is available, TD 5 training with a target network often led to better results (Table 6).

## E  Optimization

All experiments used the Adam optimiser (Kingma & Ba, 2015) with decoupled weight decay (Loshchilov & Hutter, 2017) for training. We used a weight decay scale of $10^{-4}$ and an initial learning rate of $10^{-4}$, decayed to 0 over 1 million training batches using a cosine schedule:

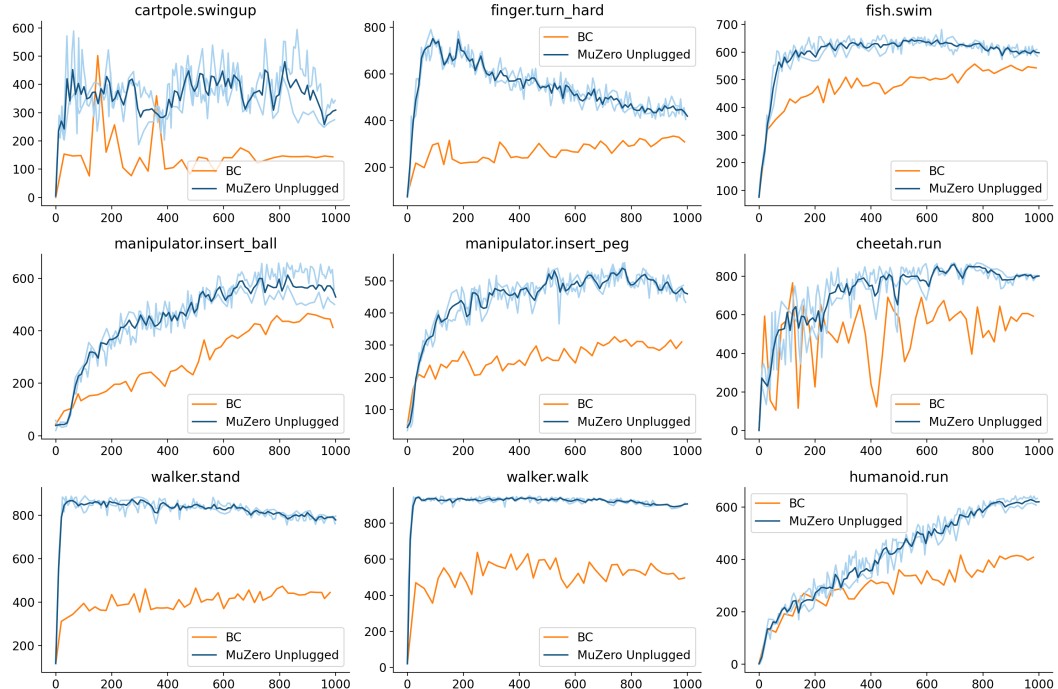

Figure 3: **Performance throughout training in RL Unplugged Control Suite.** The x-axis shows thousands of training batches, the y-axis mean reward. Performance of *MuZero Unplugged* was better than Behaviour Cloning (BC) throughout training. For some tasks with a small dataset such as cartpole, walker or finger.turn_hard, *MuZero Unplugged* performance peaked at the beginning of training and subsequently declined due to overfitting.

| Loss | supervised | | | CRR | *Reanalyse* |
|---|---|---|---|---|---|
| Unroll | 0 | 1 | 5 | 5 | 5 |
| policy | 50.7 | 49.6 | 46.9 | 271.2 | 433.0 |
| value | 143.2 | 151.2 | 359.4 | 346.7 | 549.7 |
| MCTS | - | 248.4 | 394.8 | 408.3 | 595.5 |

Table 7: **Mean score in RL Unplugged Atari: ablations of action selection and training loss**. As Table 2, but showing mean normalized score instead of median normalized score.

$$\mathrm{lr} = \mathrm{lr_{init}} \frac{1 + \cos \pi \frac{\mathrm{step}}{\mathrm{max\_steps}}}{2}$$

where $\mathrm{lr_{init}} = 10^{-4}$ and $\mathrm{max\_steps} = 10^6$.

The batch size was $1024$ for all experiments.

# F  Evaluation

Unless otherwise noted, all results were obtained by evaluating the final network checkpoint, at the end of training for 1 million mini-batches. Results are reported as the mean for 300 evaluation episodes.

In domains with continuous action spaces, we followed previous work (Gulcehre et al., 2020) in reducing the scale of the Gaussian policy close to 0 to obtain the performance of the policy mode. However, since setting the scale to 0 would not allow us to sample a set of different actions for MCTS to consider, we instead use a softer approach: $\mathrm{scale_{eval}} = \min(\mathrm{scale_{predicted}}, 0.05)$.

# G Other Hyperparameters

Our hyperparameters follow previous work (Schrittwieser et al., 2020; Hubert et al., 2021), but we reproduce them here for convenience.

We used a discount of $0.997$ for Atari and $0.99$ for DM Control experiments.

For replay, we kept a buffer of the most recent $50000$ subsequences in Atari and $2000$ in DM Control, splitting episodes into subsequences of length up to $500$. Samples were drawn from the replay buffer according to prioritised replay (Schaul et al., 2016), with priority $P(i) = \frac{p_i^\alpha}{\sum_k p_k^\alpha}$, where $p_i = |\nu_i - z_i|$, $\nu$ is the search value and $z$ the observed n-step return. To correct for sampling bias introduced by the prioritised sampling, we scaled the loss using the importance sampling ratio $w_i = (\frac{1}{N} \cdot \frac{1}{P(i)})^\beta$. In all our experiments, we set $\alpha = \beta = 1$.

# H Computational Resources

All experiments were run using third generation Google Cloud TPUs (Google, 2018).

For each game in Atari, we used 8 TPUs for training and 4 TPUs for acting and/or reanalyse for approximately 12 hours - equivalent to 1 week on a single V100 GPU.

For each level in the DM Control suite we used 1 TPU for training and 4 TPUs for acting and/or reanalyse for approximately 12 hours, equivalent to roughly 2.5 days on a single V100 GPU. Resource usage is much lower than in Atari since observations are feature instead of image based, allowing for a smaller fully-connected neural network. The larger proportion of acting to training TPUs is primarily due to inefficiencies in our implementation, which was originally optimised for larger neural networks; a more optimized implementation could reduce this further.

# I Pseudocode

We also provide a detailed description of *MuZero Unplugged* in the form of pseudocode at the end of this pdf.

| Task | # dims | no inject | inject |
|---|---|---|---|
| fish.swim | 5 | 79.7 | 585.4 |
| manipulator.insert_ball | 5 | 21.2 | 557.0 |
| manipulator.insert_peg | 5 | 90.1 | 432.7 |
| walker.stand | 6 | 735.6 | 759.8 |
| cheetah.run | 6 | 101.3 | 798.9 |
| humanoid.run | 21 | 3.3 | 633.4 |

Table 8: **Impact of injecting the trajectory action when reanalysing**. When reanalysing continuous action offline data or demonstrations from other agents, the learned policy is unlikely to sample the same actions as occur in the data, preventing the MCTS from considering those actions. This effect is especially pronounced in high dimensional tasks. Injecting the trajectory actions as one of the actions for MCTS to consider avoids this issue.

| Task | # dims | # episodes | # hidden size | | | | |
|---|---|---|---|---|---|---|---|
| | | | 32 | 64 | 128 | 256 | 512 |
| cartpole.swingup | 1 | 40 | 605.0 | 343.3 | 293.2 | 302.6 | 259.3 |
| fish.swim | 5 | 200 | 511.9 | 566.0 | 579.1 | 614.7 | 458.2 |
| walker.stand | 6 | 200 | 820.2 | 889.4 | 868.4 | 782.6 | 676.1 |
| walker.walk | 6 | 200 | 922.8 | 954.0 | 920.6 | 919.9 | 904.5 |

Table 9: **Varying network size in DM Control benchmark from RL Unplugged**. In tasks with very small amounts of training data, networks with a large number of parameters overfit easily.

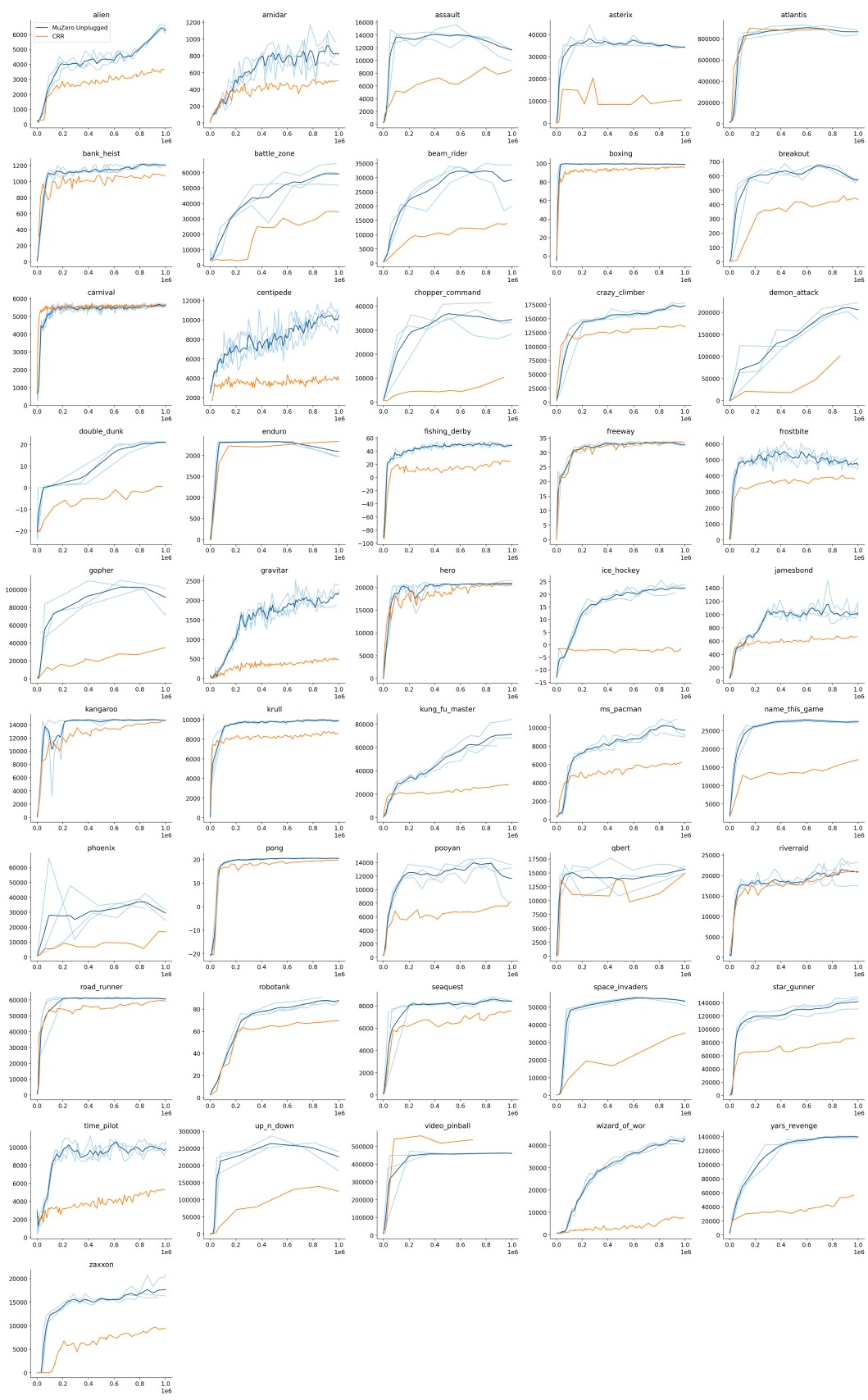

Figure 4: **Performance throughout training in RL Unplugged Atari.** The x-axis shows thousands of training batches, the y-axis mean reward. Performance of three seeds for *MuZero Unplugged* with CRR shown as a baseline.

| Game | Random | Human | *MuZero* no-op starts | | *MuZero* sticky actions | | *MuZero* Res2 Adam | |
|---|---|---|---|---|---|---|---|---|
| | | | mean | normalized | mean | normalized | mean | normalized |
| alien | 227.75 | 7,127.80 | **84,300.22** | 1,218.4 % | 56,834.58 | 820.4 % | 70,192.35 | 1,014.0 % |
| amidar | 5.77 | 1,719.53 | **13,011.88** | 758.9 % | 1,516.53 | 88.2 % | 1,197.38 | 69.5 % |
| assault | 222.39 | 742.00 | 38,745.37 | 7,413.8 % | **42,742.00** | 8,183.0 % | 33,292.22 | 6,364.4 % |
| asterix | 210.00 | 8,503.33 | 860,812.50 | 10,377.0 % | **879,375.00** | 10,600.9 % | 862,406.65 | 10,396.3 % |
| asteroids | 719.10 | 47,388.67 | 265,336.41 | 567.0 % | 374,146.38 | 800.2 % | **476,412.00** | 1,019.3 % |
| atlantis | 12,850.00 | 29,028.13 | 1,055,658.62 | 6,445.8 % | **1,353,616.62** | 8,287.5 % | 1,137,475.12 | 6,951.5 % |
| bank heist | 14.20 | 753.13 | 1,429.19 | 191.5 % | 1,077.31 | 143.9 % | **27,219.80** | 3,681.8 % |
| battle zone | 2,360.00 | 37,187.50 | 148,691.02 | 420.2 % | 167,411.77 | 473.9 % | **178,716.90** | 506.4 % |
| beam rider | 363.88 | 16,926.53 | 125,085.20 | 753.0 % | 201,154.00 | 1,212.3 % | **333,077.44** | 2,008.8 % |
| berzerk | 123.65 | 2,630.42 | **5,821.18** | 227.3 % | 1,698.21 | 62.8 % | 2,705.82 | 103.0 % |
| bowling | 23.11 | 160.73 | 138.53 | 83.7 % | 133.23 | 80.0 % | 131.65 | 78.9 % |
| boxing | 0.05 | 12.06 | 99.99 | 832.1 % | 99.96 | 831.9 % | **100.00** | 832.2 % |
| breakout | 1.72 | 30.47 | **851.92** | 2,957.2 % | 798.75 | 2,772.3 % | 758.04 | 2,630.7 % |
| centipede | 2,090.87 | 12,017.04 | 660,348.19 | 6,631.5 % | 774,420.88 | 7,780.7 % | **874,301.64** | 8,787.0 % |
| chopper command | 811.00 | 7,387.80 | **961,926.94** | 14,613.7 % | 8,945.45 | 123.7 % | 5,989.55 | 78.7 % |
| crazy climber | 10,780.50 | 35,829.41 | **196,566.92** | 741.7 % | 184,394.12 | 693.1 % | 158,541.58 | 589.9 % |
| defender | 2,874.50 | 18,688.89 | 519,002.88 | 3,263.7 % | 554,491.69 | 3,488.1 % | **557,200.75** | 3,505.2 % |
| demon attack | 152.07 | 1,971.00 | 141,508.56 | 7,771.4 % | 142,509.17 | 7,826.4 % | **143,838.04** | 7,899.5 % |
| double dunk | -18.55 | -16.40 | **23.92** | 1,975.3 % | 23.46 | 1,954.0 % | 23.91 | 1,974.9 % |
| enduro | 0.00 | 860.53 | 0.00 | 0.0 % | **2,369.00** | 275.3 % | 2,365.81 | 274.9 % |
| fishing derby | -91.71 | -38.80 | 62.54 | 291.5 % | 57.70 | 282.4 % | **73.94** | 313.1 % |
| freeway | 0.01 | 29.60 | 30.57 | 103.3 % | 0.00 | -0.0 % | **33.87** | 114.4 % |
| frostbite | 65.20 | 4,334.67 | 30,225.09 | 706.4 % | 17,087.18 | 398.7 % | **374,769.76** | 8,776.4 % |
| gopher | 257.60 | 2,412.50 | 78,132.26 | 3,613.8 % | 122,025.00 | 5,650.7 % | **122,882.50** | 5,690.5 % |
| gravitar | 173.00 | 3,351.43 | **12,709.44** | 394.4 % | 10,301.37 | 318.7 % | 8,006.93 | 246.5 % |
| hero | 1,026.97 | 30,826.38 | 36,592.23 | 119.3 % | 36,062.90 | 117.6 % | **37,234.31** | 121.5 % |
| ice hockey | -11.15 | 0.88 | 14.62 | 214.2 % | 26.26 | 311.0 % | **41.66** | 439.0 % |
| jamesbond | 29.00 | 302.80 | **33,799.58** | 12,334.0 % | 14,871.88 | 5,421.1 % | 28,626.23 | 10,444.6 % |
| kangaroo | 52.00 | 3,035.00 | **14,556.96** | 486.3 % | 14,380.00 | 480.3 % | 13,838.00 | 462.2 % |
| krull | 1,598.05 | 2,665.53 | 12,537.17 | 1,024.8 % | 11,476.19 | 925.4 % | **72,570.50** | 6,648.6 % |
| kung fu master | 258.50 | 22,736.25 | **164,599.47** | 731.1 % | 148,935.72 | 661.4 % | 116,726.96 | 518.1 % |
| montezuma revenge | 0.00 | 4,753.33 | 0.00 | 0.0 % | 0.00 | 0.0 % | 2,500.00 | 52.6 % |
| ms pacman | 307.30 | 6,951.60 | 59,421.84 | 889.7 % | 51,310.00 | 767.6 % | **70,659.76** | 1,058.8 % |
| name this game | 2,292.35 | 8,049.00 | **101,463.46** | 1,722.7 % | 85,331.43 | 1,442.5 % | 101,197.71 | 1,718.1 % |
| phoenix | 761.40 | 7,242.60 | 121,017.59 | 1,855.5 % | 105,592.66 | 1,617.5 % | **815,728.70** | 12,574.3 % |
| pitfall | -229.44 | 6,463.69 | 0.00 | 3.4 % | 0.00 | 3.4 % | 0.00 | 3.4 % |
| pong | -20.71 | 14.59 | 20.93 | 118.0 % | 20.93 | 118.0 % | **20.95** | 118.0 % |
| private eye | 24.94 | 69,571.27 | 200.00 | 0.3 % | 100.00 | 0.1 % | 100.00 | 0.1 % |
| qbert | 163.88 | 13,455.00 | 62,678.37 | 470.3 % | **102,129.41** | 767.2 % | 94,906.25 | 712.8 % |
| riverraid | 1,338.50 | 17,118.00 | **173,779.34** | 1,092.8 % | 137,983.33 | 866.0 % | 171,673.78 | 1,079.5 % |
| road runner | 11.50 | 7,845.00 | 380,115.75 | 4,852.3 % | **604,083.31** | 7,711.4 % | 531,097.00 | 6,779.7 % |
| robotank | 2.16 | 11.94 | 66.46 | 657.5 % | 69.93 | 692.9 % | **100.59** | 1,006.4 % |
| seaquest | 68.40 | 42,054.71 | 352,721.81 | 839.9 % | 399,763.62 | 952.0 % | **999,659.18** | 2,380.8 % |
| skiing | -17,098.09 | -4,336.93 | -28,449.98 | -89.0 % | -30,000.00 | -101.1 % | -30,000.00 | -101.1 % |
| solaris | 1,236.30 | 12,326.67 | 3,507.99 | 20.5 % | 5,860.00 | 41.7 % | 5,132.95 | 35.1 % |
| space invaders | 148.03 | 1,668.67 | **3,663.32** | 231.2 % | 3,639.04 | 229.6 % | 3,645.63 | 230.0 % |
| star gunner | 664.00 | 10,250.00 | **156,559.81** | 1,626.3 % | 127,416.66 | 1,322.3 % | 154,548.26 | 1,605.3 % |
| surround | -9.99 | 6.53 | 8.82 | 113.9 % | 8.62 | 112.7 % | **9.90** | 120.4 % |
| tennis | -23.84 | -8.27 | -0.23 | 151.6 % | **0.00** | 153.1 % | -0.00 | 153.1 % |
| time pilot | 3,568.00 | 5,229.10 | 183,259.78 | 10,817.6 % | **427,209.09** | 25,503.6 % | 424,011.16 | 25,311.1 % |
| tutankham | 11.43 | 167.59 | 290.98 | 179.0 % | 235.00 | 143.2 % | **347.99** | 215.5 % |
| up n down | 533.40 | 11,693.23 | 441,300.81 | 3,949.6 % | 522,961.53 | 4,681.3 % | **634,898.18** | 5,684.4 % |
| venture | 0.00 | 1,187.50 | 0.00 | 0.0 % | 0.00 | 0.0 % | **1,731.47** | 145.8 % |
| video pinball | 0.00 | 17,667.90 | 638,373.31 | 3,613.2 % | 775,303.81 | 4,388.2 % | **865,543.44** | 4,899.0 % |
| wizard of wor | 563.50 | 4,756.52 | **104,527.07** | 2,479.4 % | 0.00 | -13.4 % | 100,096.60 | 2,373.8 % |
| yars revenge | 3,092.91 | 54,576.93 | 829,662.62 | 1,605.5 % | **846,060.69** | 1,637.3 % | 219,838.09 | 421.0 % |
| zaxxon | 32.50 | 9,173.30 | 0.53 | -0.3 % | 58,115.00 | 635.4 % | **154,131.86** | 1,685.8 % |
| # best | 0 | 6 | 16 | | 9 | | 26 | |
| median | | | | 741.7 % | | 692.9 % | | 1,006.4 % |
| mean | | | | 2,183.6 % | | 2,188.4 % | | 2,856.2 % |

Table 10: **Evaluation of *MuZero* in Atari for individual games.** Best result for each game highlighted in **bold**. Each episode is limited to a maximum of 30 minutes of game time (108k frames). Human normalized score is calculated as $s_{\text{normalized}} = \frac{s_{\text{agent}} - s_{\text{random}}}{s_{\text{human}} - s_{\text{random}}}$. The original *MuZero* was trained without sticky actions and evaluated with 30 random no-op starts, we reproduce the results from (Schrittwieser et al., 2020) here for easy reference. Our version of *MuZero* was trained and evaluated with *sticky actions* (Machado et al., 2017). As shown in the table, this does not negatively impact performance, mean and median score are essentially unchanged. Finally, our version of *MuZero* with using Res v2 (He et al., 2016) and trained with the Adam optimiser shows clear improvements.

| Game | Random | Online DQN | BC | DQN | IQN | BCQ | REM | policy | supervised CRR | max-v | MCTS | *MuZero* Unplugged |
|---|---|---|---|---|---|---|---|---|---|---|---|---|
| alien | 199.8 | 2766.8 | 2670.0 | 1690.0 | 2860.0 | 2090.0 | 1730.0 | 1621.0 | 3658.0 | 3414.7 | 3699.7 | **6335.4** |
| amidar | 3.2 | **1557.0** | 256.0 | 224.0 | 351.0 | 254.0 | 214.0 | 157.7 | 499.3 | 399.4 | 540.7 | 962.7 |
| assault | 235.2 | 1946.1 | 1810.0 | 1940.0 | 2180.0 | 2260.0 | 3070.0 | 1461.3 | 8537.5 | 11063.5 | **13517.6** | 13406.9 |
| asterix | 279.1 | 4131.8 | 2960.0 | 1520.0 | 5710.0 | 1930.0 | 4890.0 | 2742.2 | 10484.5 | 28217.5 | 24016.0 | **34674.0** |
| atlantis | 16973.0 | 944228.0 | 2390000.0 | 3020000.0 | 2710000.0 | 3200000.0 | **3360000.0** | 175194.0 | 888496.5 | 399428.5 | 893635.5 | 873651.5 |
| bank heist | 13.7 | 907.7 | 1050.0 | 50.0 | 1110.0 | 270.0 | 160.0 | 619.0 | 1071.7 | 1057.2 | **1210.3** | 1168.3 |
| battle zone | 2786.8 | 26459.0 | 4800.0 | 25600.0 | 16500.0 | 25400.0 | 26200.0 | 15180.0 | 34455.0 | 33290.0 | 43375.0 | **65815.0** |
| beam rider | 362.1 | 6453.3 | 1480.0 | 1810.0 | 3020.0 | 1990.0 | 2200.0 | 2418.0 | 13965.5 | 15268.1 | 18499.7 | **33150.9** |
| boxing | 0.8 | 84.1 | 83.9 | 96.3 | 95.8 | 97.2 | 97.3 | 74.9 | 95.6 | 97.7 | 95.8 | **99.4** |
| breakout | 1.3 | 157.9 | 235.0 | 324.0 | 314.0 | 375.0 | 362.0 | 97.8 | 435.1 | 458.6 | 503.8 | **582.2** |
| carnival | 669.5 | 5339.5 | 3920.0 | 1450.0 | 4820.0 | 4310.0 | 2080.0 | 5160.7 | 5588.8 | 4945.8 | 5525.2 | **5609.5** |
| centipede | 2181.7 | 3972.5 | 1070.0 | 1250.0 | 1830.0 | 1430.0 | 810.0 | 2304.8 | 3953.9 | 4460.7 | 4397.6 | **10764.0** |
| chopper command | 823.1 | 3678.2 | 660.0 | 2250.0 | 830.0 | 3950.0 | 3610.0 | 1409.0 | 10315.0 | 15332.0 | 16929.5 | **41567.0** |
| crazy climber | 8173.6 | 118080.2 | 123000.0 | 23000.0 | 126000.0 | 28000.0 | 42000.0 | 94845.5 | 135524.5 | 140857.0 | 150201.5 | **167372.5** |
| demon attack | 166.0 | 6517.0 | 7600.0 | 11000.0 | 15500.0 | 19300.0 | 17000.0 | 2984.0 | 101626.9 | 101440.5 | 118643.1 | **212935.6** |
| double dunk | -18.4 | -1.2 | -16.4 | -17.9 | -16.7 | -12.9 | -17.9 | -16.9 | 0.5 | -3.0 | -0.8 | **20.9** |
| enduro | 0.0 | 1016.3 | 720.0 | 1210.0 | 1700.0 | 1390.0 | **3650.0** | 534.2 | 2334.6 | 1830.1 | 2193.8 | 2334.2 |
| fishing derby | -93.2 | 18.6 | -7.4 | 17.0 | 20.8 | 28.9 | 29.3 | -6.1 | 24.8 | 14.1 | 11.7 | **46.0** |
| freeway | 0.0 | 26.8 | 21.8 | 15.4 | 24.7 | 16.9 | 7.2 | 28.0 | **33.7** | 27.6 | 33.1 | 32.4 |
| frostbite | 72.0 | 1643.7 | 780.0 | 3230.0 | 2630.0 | 3520.0 | 3070.0 | 828.6 | 3812.0 | 4518.3 | 4377.4 | **5094.9** |
| gopher | 282.6 | 8241.0 | 4900.0 | 2400.0 | 11300.0 | 8700.0 | 3700.0 | 3854.7 | 34731.1 | 93322.6 | 83799.2 | **101380.3** |
| gravitar | 213.7 | 310.6 | 20.0 | 500.0 | 235.0 | 580.0 | 424.0 | 229.8 | 479.5 | 648.2 | 709.2 | **1891.6** |
| hero | 719.1 | 16233.5 | 13900.0 | 5200.0 | 16200.0 | 13200.0 | 14000.0 | 11251.6 | 20615.4 | 20485.1 | 20755.3 | **20757.1** |
| ice hockey | -9.8 | -4.0 | -5.6 | -2.9 | -4.7 | -2.5 | -1.2 | -6.3 | -1.5 | -4.3 | -4.0 | **20.7** |
| jamesbond | 27.6 | 777.7 | 237.0 | 490.0 | 699.0 | 438.0 | 369.0 | 393.0 | 670.8 | 649.2 | 753.5 | **945.2** |
| kangaroo | 41.3 | 14125.1 | 5690.0 | 820.0 | 9120.0 | 1300.0 | 1210.0 | 4701.0 | 14463.0 | 5817.0 | 12045.5 | **14736.0** |
| krull | 1556.9 | 7238.5 | 8500.0 | 7480.0 | 8470.0 | 7780.0 | 7980.0 | 5835.9 | 8624.2 | 8803.3 | 9347.3 | **9815.3** |
| kung fu master | 556.3 | 26637.9 | 5100.0 | 16100.0 | 19500.0 | 16900.0 | 19400.0 | 8998.0 | 28020.0 | 24526.0 | 39628.5 | **61354.0** |
| ms pacman | 248.0 | 4171.5 | 4040.0 | 2470.0 | 4390.0 | 3080.0 | 3150.0 | 2946.8 | 6240.7 | 6745.7 | 6839.1 | **10934.1** |
| name this game | 2401.1 | 8645.1 | 4100.0 | 11500.0 | 9900.0 | 12600.0 | 13000.0 | 6043.2 | 17094.8 | 19084.8 | 19090.2 | **27429.3** |
| phoenix | 873.0 | 5122.3 | 2940.0 | 6410.0 | 4940.0 | 6620.0 | 7480.0 | 4376.6 | 16698.7 | 37443.7 | **38667.5** | 32532.8 |
| pong | -20.3 | 18.2 | 18.9 | 12.9 | 19.2 | 16.5 | 16.5 | 15.0 | 19.7 | 2.7 | 14.5 | **20.6** |
| pooyan | 411.4 | 4135.3 | 3850.0 | 3180.0 | 5000.0 | 4200.0 | 4470.0 | 2703.8 | 8082.8 | 7098.6 | 8916.2 | **13630.0** |
| qbert | 155.0 | 12275.1 | 12600.0 | 10600.0 | 13400.0 | 12600.0 | 13100.0 | 8296.6 | 14900.6 | 12469.9 | 13428.0 | **14943.2** |
| riverraid | 1504.2 | 12798.9 | 6000.0 | 9100.0 | 13000.0 | 14200.0 | 14200.0 | 8476.6 | 21091.2 | 20142.1 | 21509.6 | **22245.7** |
| road runner | 15.5 | 47880.5 | 19000.0 | 31700.0 | 44700.0 | 57400.0 | 56500.0 | 31321.5 | 59337.5 | 59514.0 | **60856.0** | 60465.0 |
| robotank | 2.0 | 63.4 | 15.7 | 55.7 | 42.7 | 60.7 | 60.5 | 27.6 | 69.4 | 74.5 | 75.2 | **91.1** |
| seaquest | 81.8 | 3233.5 | 150.0 | 2870.0 | 1670.0 | 5410.0 | 5910.0 | 802.7 | 7514.2 | 7315.1 | 7714.5 | **8411.8** |
| space invaders | 149.5 | 2044.6 | 790.0 | 2710.0 | 2840.0 | 2920.0 | 2810.0 | 1235.5 | 35375.9 | 45017.3 | 46391.2 | **55704.7** |
| star gunner | 677.2 | 55103.8 | 3000.0 | 1600.0 | 39400.0 | 2500.0 | 7500.0 | 11661.0 | 86955.0 | 98493.0 | 101165.0 | **145711.5** |
| time pilot | 3450.9 | 4160.5 | 1950.0 | 5310.0 | 3140.0 | 5180.0 | 4490.0 | 2157.5 | 5282.5 | 6572.5 | 7786.0 | **9427.5** |
| up n down | 513.9 | 15677.9 | 16300.0 | 14600.0 | 32300.0 | 32500.0 | 27600.0 | 6127.2 | 124783.7 | 209816.5 | 244367.5 | **251927.9** |
| video pinball | 26024.4 | 335055.7 | 27000.0 | 82000.0 | 102000.0 | 103000.0 | 313000.0 | 100873.0 | **537485.5** | 500181.8 | 465922.8 | 462317.5 |
| wizard of wor | 686.6 | 1787.8 | 730.0 | 2300.0 | 1400.0 | 4680.0 | 2730.0 | 846.0 | 7531.0 | 15541.0 | 18634.5 | **40651.0** |
| yars revenge | 3147.7 | 26763.0 | 19100.0 | 24900.0 | 28400.0 | 29100.0 | 23100.0 | 20572.5 | 56552.6 | 76380.4 | 77773.8 | **141317.3** |
| zaxxon | 10.6 | 4681.9 | 10.0 | 6050.0 | 870.0 | 9430.0 | 8300.0 | 2457.0 | 9397.0 | 11864.0 | 12102.5 | **16143.5** |

Table 11: **Evaluation of *MuZero* in RL Unplugged for individual games.** Best result for each game highlighted in **bold**. Each episode is limited to a maximum of 30 minutes of game time (108k frames).

```python
"""Pseudocode description of the MuZero algorithm."""
# pylint: disable=unused-argument
# pylint: disable=missing-docstring
# pylint: disable=g-explicit-length-test

import collections
import math
import random
import typing
from typing import Dict, List, Optional

import dataclasses
import haiku as hk
import jax
import jax.numpy as jnp
import numpy
import optax

##########################
####### Helpers ##########

MAXIMUM_FLOAT_VALUE = float('inf')

KnownBounds = collections.namedtuple('KnownBounds', ['min', 'max'])

class MinMaxStats(object):
  """A class that holds the min-max values of the tree."""

  def __init__(self):
    self.maximum = -MAXIMUM_FLOAT_VALUE
    self.minimum = MAXIMUM_FLOAT_VALUE

  def update(self, value: float):
    self.maximum = max(self.maximum, value)
    self.minimum = min(self.minimum, value)

  def normalize(self, value: float) -> float:
    if self.maximum > self.minimum:
      # We normalize only when we have set the maximum and minimum values.
      return (value - self.minimum) / (self.maximum - self.minimum)
    return value

def visit_softmax_temperature(num_moves, training_steps):
  if training_steps < 500e3:
    return 1.0
  elif training_steps < 750e3:
    return 0.5
  else:
    return 0.25

class MuZeroConfig(object):
```

```python
    def __init__(self, action_space_size: int, max_moves: int, discount: float,
                 dirichlet_alpha: float, num_simulations: int, td_steps: int,
                 num_actors: int):
        ### Self-Play
        self.action_space_size = action_space_size
        self.num_actors = num_actors

        self.visit_softmax_temperature_fn = visit_softmax_temperature
        self.max_moves = max_moves
        self.num_simulations = num_simulations
        self.discount = discount

        # Root prior exploration noise.
        self.root_dirichlet_alpha = dirichlet_alpha
        self.root_exploration_fraction = 0.25

        # UCB formula
        self.pb_c_base = 19652
        self.pb_c_init = 1.25

        ### Training
        self.training_steps = int(1000e3)
        self.checkpoint_interval = 500
        self.window_size = int(1e6)
        self.batch_size = 1024
        self.num_unroll_steps = 5
        self.td_steps = td_steps

        # Cosine learning rate schedule with decoupled weight decay.
        self.lr_init = 1e-4
        self.weight_decay = 1e-4

        # Reanalyse fraction controls the trade-off between environment interactions
        # and running search on stored data.
        # 0.0 corresponds to only environment intercations, 1.0 to the full offline
        # case.
        self.reanalyse_fraction = 1.0

    def new_game(self):
        return Game(self.action_space_size, self.discount, Environment())

def make_atari_config() -> MuZeroConfig:

    return MuZeroConfig(
        action_space_size=18,
        max_moves=27000,  # Half an hour at action repeat 4.
        discount=0.997,
        dirichlet_alpha=0.25,
        num_simulations=50,
        td_steps=5,
        num_actors=350)
```

```python
class Action(object):

  def __init__(self, index: int):
    self.index = index

  def __hash__(self):
    return self.index

  def __eq__(self, other):
    return self.index == other.index

  def __gt__(self, other):
    return self.index > other.index

class Player(object):
  pass

class Node(object):

  def __init__(self, prior: float):
    self.visit_count = 0
    self.to_play = -1
    self.prior = prior
    self.value_sum = 0
    self.children = {}
    self.hidden_state = None
    self.reward = 0

  def expanded(self) -> bool:
    return len(self.children) > 0

  def value(self) -> float:
    if self.visit_count == 0:
      return 0
    return self.value_sum / self.visit_count

class ActionHistory(object):
  """Simple history container used inside the search.

  Only used to keep track of the actions executed.
  """

  def __init__(self, history: List[Action], action_space_size: int):
    self.history = list(history)
    self.action_space_size = action_space_size

  def clone(self):
    return ActionHistory(self.history, self.action_space_size)

  def add_action(self, action: Action):
    self.history.append(action)
```

```python
  def last_action(self) -> Action:
    return self.history[-1]

  def action_space(self) -> List[Action]:
    return [Action(i) for i in range(self.action_space_size)]

  def to_play(self) -> Player:
    return Player()

class Environment(object):
  """The environment MuZero is interacting with."""

  def step(self, action) -> float:
    return 0

class Game(object):
  """A single episode of interaction with the environment."""

  def __init__(self, action_space_size: int, discount: float,
               env: Optional[Environment]):
    self.environment = env
    self.history = []
    self.rewards = []
    self.child_visits = []
    self.root_values = []
    self.action_space_size = action_space_size
    self.discount = discount

  def terminal(self) -> bool:
    # Game specific termination rules.
    pass

  def legal_actions(self) -> List[Action]:
    # Game specific calculation of legal actions.
    return []

  def apply(self, action: Action):
    if self.environment:
      reward = self.environment.step(action)
      self.rewards.append(reward)
    self.history.append(action)

  def store_search_statistics(self, root: Node):
    sum_visits = sum(child.visit_count for child in root.children.values())
    action_space = (Action(index) for index in range(self.action_space_size))
    self.child_visits.append([
        root.children[a].visit_count / sum_visits if a in root.children else 0
        for a in action_space
    ])
    self.root_values.append(root.value())

  def make_image(self, state_index: int):
    # Game specific feature planes.
```

```python
      return []

  def make_target(self, state_index: int, num_unroll_steps: int, td_steps: int,
                  to_play: Player):
    # The value target is the discounted root value of the search tree N steps
    # into the future, plus the discounted sum of all rewards until then.
    targets = []
    for current_index in range(state_index, state_index + num_unroll_steps + 1):
      bootstrap_index = current_index + td_steps
      if bootstrap_index < len(self.root_values):
        value = self.root_values[bootstrap_index] * self.discount**td_steps
      else:
        value = 0

      for i, reward in enumerate(self.rewards[current_index:bootstrap_index]):
        value += reward * self.discount**i  # pytype: disable=unsupported-operands

      if current_index > 0 and current_index <= len(self.rewards):
        last_reward = self.rewards[current_index - 1]
      else:
        last_reward = None

      if current_index < len(self.root_values):
        targets.append((value, last_reward, self.child_visits[current_index]))
      else:
        # States past the end of games are treated as absorbing states.
        targets.append((0, last_reward, []))
    return targets

  def to_play(self) -> Player:
    return Player()

  def action_history(self) -> ActionHistory:
    return ActionHistory(self.history, self.action_space_size)

  def is_reanalyse(self) -> bool:
    return self.environment is None

class StoredGame(Game):
  """A stored Game that can be used for reanalyse."""

  def __init__(self, game: Game):
    super().__init__(game.action_space_size, game.discount, env=None)
    self._stored_history = game.history
    self._stored_rewards = game.rewards

  def terminal(self) -> bool:
    return not self._stored_history

  def apply(self, action: Action):
    # Ignore the action, instead replay the stored data.
    del action
    self.rewards.append(self._stored_rewards.pop(0))
    self.history.append(self._stored_history.pop(0))
```

```python
class ReplayBuffer(object):

  def __init__(self, config: MuZeroConfig):
    self.window_size = config.window_size
    self.batch_size = config.batch_size
    self.buffer = []

  def save_game(self, game):
    if len(self.buffer) > self.window_size:
      self.buffer.pop(0)
    self.buffer.append(game)

  def sample_batch(self, num_unroll_steps: int, td_steps: int):
    games = [self.sample_game() for _ in range(self.batch_size)]
    game_pos = [(g, self.sample_position(g)) for g in games]
    return [(g.make_image(i), g.history[i:i + num_unroll_steps],
             g.make_target(i, num_unroll_steps, td_steps, g.to_play()))
            for (g, i) in game_pos]

  def sample_game(self) -> Game:
    # Sample game from buffer either uniformly or according to some priority.
    return self.buffer[0]

  def sample_position(self, game) -> int:
    # Sample position from game either uniformly or according to some priority.
    return -1

class ReanalyseBuffer(object):

  def __init__(self):
    self._games = []

  def save_game(self, game: Game):
    """Saves a new game to the reanalyse buffer, to be reanalysed later."""

  def sample_game(self) -> Game:
    """Samples a game that should be reanalysed."""
    games = random.choices(
        self._games, weights=[len(g.history) for g in self._games], k=1)
    return StoredGame(games[0])

class DemonstrationBuffer(ReanalyseBuffer):
  """A reanlayse buffer of a fixed set of demonstrations.

  Can be used to learn from existing policies, human demonstrations or for
  Offline RL.
  """

  def __init__(self, demonstrations: List[Game]):
    super().__init__()
    self._games.extend(demonstrations)
```

```python
class MostRecentBuffer(ReanalyseBuffer):
  """A reanalyse buffer that keeps the most recent games to reanalyse."""

  def __init__(self, capacity: int):
    super().__init__()
    self._capacity = capacity

  def save_game(self, game: Game):
    self._games.append(game)
    while sum(len(g.history) for g in self._games):
      self._games.pop(0)

class HighestRewardBuffer(ReanalyseBuffer):
  """A reanalyse buffer that keeps games with highest rewards to reanalyse."""

  def __init__(self, capacity: int):
    super().__init__()
    self._capacity = capacity

  def save_game(self, game: Game):
    self._games.append(game)
    while sum(len(g.history) for g in self._games):
      self._games.sort(key=lambda g: sum(g.rewards), reverse=True)
      self._games.pop()

class NetworkOutput(typing.NamedTuple):
  value: float
  reward: float
  policy_logits: Dict[Action, float]
  hidden_state: List[float]

class Network(object):

  def __init__(self,
               params: Optional[hk.Params] = None,
               state: Optional[hk.State] = None):
    pass

  def initial_inference(self, image) -> NetworkOutput:
    # representation + prediction function
    return NetworkOutput(0, 0, {}, [])

  def recurrent_inference(self, hidden_state, action) -> NetworkOutput:
    # dynamics + prediction function
    return NetworkOutput(0, 0, {}, [])

  def get_weights(self):
    # Returns the weights of this network.
    return []
```

```python
  def training_steps(self) -> int:
    # How many steps / batches the network has been trained for.
    return 0

class SharedStorage(object):

  def __init__(self):
    self._networks = {}

  def latest_network(self) -> Network:
    if self._networks:
      return self._networks[max(self._networks.keys())]
    else:
      # policy -> uniform, value -> 0, reward -> 0
      return make_uniform_network()

  def save_network(self, step: int, network: Network):
    self._networks[step] = network

def bernoulli(p: float) -> bool:
  return random.random() < p

@dataclasses.dataclass
class GenerationStats:
  total_states: int = 0
  max_episode_length: int = 0
  mean_episode_length: Optional[float] = None

  def episode_length(self):
    if self.mean_episode_length is None:
      return self.max_episode_length
    else:
      return self.mean_episode_length

class ActingStats(object):

  def __init__(self, config: MuZeroConfig):
    self._reanalyse_fraction = config.reanalyse_fraction
    self._fresh = GenerationStats()
    self._reanalysed = GenerationStats()

  def state_added(self, game: Game):
    stats = self._get_stats(game.is_reanalyse())
    stats.total_states += 1
    stats.max_episode_length = max(stats.max_episode_length, len(game.history))

  def game_finished(self, game: Game):
    stats = self._get_stats(game.is_reanalyse())
    if stats.mean_episode_length is None:
      stats.mean_episode_length = len(game.history)
    else:
```

```
        alpha = 0.01
        stats.mean_episode_length = alpha * len(
            game.history) + (1 - alpha) * stats.mean_episode_length

    def should_reanalyse(self) -> bool:
        # Overshoot slightly to approach desired fraction.
        actual = self._reanalysed.total_states / (
            self._reanalysed.total_states + self._fresh.total_states)
        target = self._reanalyse_fraction + (self._reanalyse_fraction - actual) / 2
        target = max(0, min(1, target))

        # Correct for reanalysing only part of full episodes.
        fresh_fraction = 1 - target
        parts_per_episode = max(
            1,
            self._fresh.episode_length() / self._reanalysed.episode_length())
        fresh_fraction /= parts_per_episode

        return bernoulli(1 - fresh_fraction)

    def _get_stats(self, reanalysed: bool) -> GenerationStats:
        return self._reanalysed if reanalysed else self._fresh

##### End Helpers ########
#########################

# MuZero training is split into two independent parts: Network training and
# self-play data generation.
# These two parts only communicate by transferring the latest network checkpoint
# from the training to the self-play, and the finished games from the self-play
# to the training.
def muzero(config: MuZeroConfig):
    storage = SharedStorage()
    replay_buffer = ReplayBuffer(config)
    reanalyse_buffer = ReanalyseBuffer()

    for _ in range(config.num_actors):
        launch_job(run_selfplay, config, storage, replay_buffer, reanalyse_buffer)

    train_network(config, storage, replay_buffer)

    return storage.latest_network()

##################################
####### Part 1: Self-Play ########

# Each self-play job is independent of all others; it takes the latest network
# snapshot, produces a game and makes it available to the training job by
# writing it to a shared replay buffer.
def run_selfplay(config: MuZeroConfig, storage: SharedStorage,
                 replay_buffer: ReplayBuffer,
```

```
                  reanalyse_buffer: ReanalyseBuffer):
  stats = ActingStats(config)
  while True:
    network = storage.latest_network()

    if stats.should_reanalyse():
      game = reanalyse_buffer.sample_game()
    else:
      game = config.new_game()

    game = play_game(config, game, network, stats)

    replay_buffer.save_game(game)
    if not game.is_reanalyse():
      reanalyse_buffer.save_game(game)

# Each game is produced by starting at the initial board position, then
# repeatedly executing a Monte Carlo Tree Search to generate moves until the end
# of the game is reached.
def play_game(config: MuZeroConfig, game: Game, network: Network,
              stats: ActingStats) -> Game:

  while not game.terminal() and len(game.history) < config.max_moves:
    min_max_stats = MinMaxStats()

    # At the root of the search tree we use the representation function to
    # obtain a hidden state given the current observation.
    root = Node(0)
    current_observation = game.make_image(-1)
    network_output = network.initial_inference(current_observation)
    expand_node(root, game.to_play(), game.legal_actions(), network_output)
    backpropagate([root], network_output.value, game.to_play(), config.discount,
                  min_max_stats)
    add_exploration_noise(config, root)

    # We then run a Monte Carlo Tree Search using only action sequences and the
    # model learned by the network.
    run_mcts(config, root, game.action_history(), network, min_max_stats)
    action = select_action(config, len(game.history), root, network)
    game.apply(action)
    game.store_search_statistics(root)
    stats.state_added(game)

  stats.game_finished(game)
  return game

# Core Monte Carlo Tree Search algorithm.
# To decide on an action, we run N simulations, always starting at the root of
# the search tree and traversing the tree according to the UCB formula until we
# reach a leaf node.
def run_mcts(config: MuZeroConfig, root: Node, action_history: ActionHistory,
             network: Network, min_max_stats: MinMaxStats):
  for _ in range(config.num_simulations):
```

```python
    history = action_history.clone()
    node = root
    search_path = [node]

    while node.expanded():
      action, node = select_child(config, node, min_max_stats)
      history.add_action(action)
      search_path.append(node)

    # Inside the search tree we use the dynamics function to obtain the next
    # hidden state given an action and the previous hidden state.
    parent = search_path[-2]
    network_output = network.recurrent_inference(parent.hidden_state,
                                                 history.last_action())
    expand_node(node, history.to_play(), history.action_space(), network_output)

    backpropagate(search_path, network_output.value, history.to_play(),
                  config.discount, min_max_stats)

def select_action(config: MuZeroConfig, num_moves: int, node: Node,
                  network: Network):
  visit_counts = [
      (child.visit_count, action) for action, child in node.children.items()
  ]
  t = config.visit_softmax_temperature_fn(
      num_moves=num_moves, training_steps=network.training_steps())
  _, action = softmax_sample(visit_counts, t)
  return action

# Select the child with the highest UCB score.
def select_child(config: MuZeroConfig, node: Node,
                 min_max_stats: MinMaxStats):
  _, action, child = max(
      (ucb_score(config, node, child, min_max_stats), action,
       child) for action, child in node.children.items())
  return action, child

# The score for a node is based on its value, plus an exploration bonus based on
# the prior.
def ucb_score(config: MuZeroConfig, parent: Node, child: Node,
              min_max_stats: MinMaxStats) -> float:
  pb_c = math.log((parent.visit_count + config.pb_c_base + 1) /
                  config.pb_c_base) + config.pb_c_init
  pb_c *= math.sqrt(parent.visit_count) / (child.visit_count + 1)

  prior_score = pb_c * child.prior
  if child.visit_count > 0:
    value_score = min_max_stats.normalize(child.reward +
                                          config.discount * child.value())
  else:
    value_score = 0
  return prior_score + value_score
```

```python
# We expand a node using the value, reward and policy prediction obtained from
# the neural network.
def expand_node(node: Node, to_play: Player, actions: List[Action],
                network_output: NetworkOutput):
  node.to_play = to_play
  node.hidden_state = network_output.hidden_state
  node.reward = network_output.reward
  policy = {a: math.exp(network_output.policy_logits[a]) for a in actions}
  policy_sum = sum(policy.values())
  for action, p in policy.items():
    node.children[action] = Node(p / policy_sum)

# At the end of a simulation, we propagate the evaluation all the way up the
# tree to the root.
def backpropagate(search_path: List[Node], value: float, to_play: Player,
                  discount: float, min_max_stats: MinMaxStats):
  for node in reversed(search_path):
    node.value_sum += value if node.to_play == to_play else -value
    node.visit_count += 1
    min_max_stats.update(node.value())

    value = node.reward + discount * value

# At the start of each search, we add dirichlet noise to the prior of the root
# to encourage the search to explore new actions.
def add_exploration_noise(config: MuZeroConfig, node: Node):
  actions = list(node.children.keys())
  noise = numpy.random.dirichlet([config.root_dirichlet_alpha] * len(actions))
  frac = config.root_exploration_fraction
  for a, n in zip(actions, noise):
    node.children[a].prior = node.children[a].prior * (1 - frac) + n * frac

######### End Self-Play ##########
################################

################################
####### Part 2: Training #########

def train_network(config: MuZeroConfig, storage: SharedStorage,
                  replay_buffer: ReplayBuffer):
  network = Network()
  network = hk.transform_with_state(network)

  # Sample initial batch of random data, generated with uniform policy.
  batch = replay_buffer.sample_batch(config.num_unroll_steps, config.td_steps)
  params, state = network.init(batch)

  def learning_rate(step: int):
    lr_decay = 0.5 * (1 + jnp.cos(jnp.pi * step / config.training_steps))
```

```python
    return config.lr_init * lr_decay

  # Adam with decoupled weight decay.
  optimizer = optax.scale_by_adam(eps=1e-8)
  optimizer = optax.chain(
      optimizer,
      optax.add_decayed_weights(config.weight_decay / config.lr_init))
  optimizer = optax.chain(optimizer, optax.scale_by_schedule(learning_rate),
                          optax.scale(-1))
  opt_state = optimizer.init(params)

  for step in range(config.training_steps):
    if step % config.checkpoint_interval == 0:
      storage.save_network(step, Network(params, state))

    batch = replay_buffer.sample_batch(config.num_unroll_steps, config.td_steps)
    grads, state = jax.grad(loss_fn)(params, state, network.apply, batch)
    updates, opt_state = optimizer.update(grads, opt_state, params)
    params = optax.apply_updates(params, updates)

  storage.save_network(config.training_steps, Network(params, state))

def scale_gradient(x: jnp.ndarray, scale: float) -> jnp.ndarray:
  """Multiplies the gradient of `x` by `scale`."""

  @jax.custom_gradient
  def wrapped(x: jnp.ndarray):
    return x, lambda grad: (grad * scale,)

  return wrapped(x)

def softmax_cross_entropy(logits, labels):
  return -jnp.sum(labels * jax.nn.log_softmax(logits), axis=-1)

def loss_fn(params: hk.Params, state: hk.State, network: Network, batch):
  loss = 0
  for image, actions, targets in batch:
    # Initial step, from the real observation.
    network_output = network.initial_inference(image)
    hidden_state = network_output.hidden_state
    predictions = [(1.0, network_output)]

    # Recurrent steps, from action and previous hidden state.
```