# OpenReview forum: "Online and Offline Reinforcement Learning by Planning with a Learned Model"
_NeurIPS.cc/2021/Conference — NeurIPS 2021 Spotlight_

### Official Review · Reviewer_zAF9 · 2021-07-14

**Rating:** 8
**Confidence:** 4

**Summary:**

The submission investigates the application of MuZero to offline RL and mixed online/offline RL using the Reanalyse algorithm introduced in MuZero. It presents experimental results for Atari with different online data proportions and for the RL Unplugged benchmark.

**Limitations And Societal Impact:**

Yes

**Main Review:**

The paper is well written and easy to follow. While the submission does not make any novel insight or propose any novel algorithm beyond a more detailed look into the performance of the Reanalyse procedure introduced in MuZero, its empirical results are strong, despite not making any offline RL specific adaptions. Given the level of recent interest in offline RL, the having the performance of MuZero benchmarked is a valuable contribution. Overall, it is a strong submission, and I don't have any major criticisms.

### Small comments

Line 4 could demonstrate -> has demonstrated

Line 44 “A lot of research” informal

On line 96 and 97 v should be \nu

Is the submission suggesting that the logarithmic scaling observed in Figure 1 holds in generality across environments?

Line 215 use em dash

Line 215 Table 2a -> Table 1a

Line 271 use em dash

### Reproducibility

It would be nice if the submission could additionally include the pseudocode in the appendix as a text file to make it easier for others to work from.

**Time Spent Reviewing:**

5

---

> ### Author Response · Authors · 2021-08-10
> **Online and Offline Reinforcement Learning by Planning with a Learned Model**
>
> We will make the pseudocode available as a separate text file; we included it in the appendix for the initial submission because we could not find a way to upload a separate standalone file to OpenReview.
>
> In our experience, logarithmic scaling does hold true across "open-ended" environments, i.e. environments where the skill ceiling is still far above the capabilities of the agent. This is the case for example for Ms. Pac-Man in Atari or Go in board games, but not true for Breakout or Pong in Atari, where even simple agents can reach the maximum score.

---

### Official Review · Reviewer_WqPe · 2021-07-16

**Rating:** 7
**Confidence:** 4

**Summary:**

In this paper the authors focus on the Reanalyze algorithm introduced in MuZero [1] and show its applicability to improving sample efficiency, as well as the offline/batch RL setting with no modification to the underlying algorithm (i.e., it can be online and offline).

They show that Dyna style re-use of previous data points provides the ability to bootstrap policy and value estimations using improved versions of themselves, thus allowing for better estimation of these quantities thanks to having a latent-space model (similar to VPN [2]) to facilitate imaginary rollouts under MCTS. They show state of the art performance in several offline RL benchmarks as a result.

**Limitations And Societal Impact:**

No issues.

**Main Review:**

Pros:
* The paper is fairly well written, and whilst the core idea is relatively simple (e.g., reapply the update steps from MuZero to previously sampled trajectories), it does a good job of delving into its inner workings and investigating how it compares to some existing approaches/losses that don’t inherently need a model (e.g., BC and CRR).
* The results seem relatively superlative, especially in the offline Atari experiments, where I believe the results are significantly better than those reported previously. Similarly, the DM Control performance appears when taken as a whole.
* The overall approach is transparent, and the authors have been clear regarding limitations as well as their experiment methodology. For instance, regarding the performance figures shown, they make it clear when they are cherry-picking offline checkpoints for online performance in their tables, and provide full training curves in the appendix to show the degradation phenomenon that can occur with increased training. Whilst this may seem like a minor point, in my experience these details often made very opaque in offline RL (for the obvious reasons), so it is refreshing to see the opposite here. Other examples are the extensive pseudocode in the appendix, as well as baselining a BC version of their MuZero against an explicit BC model to affirm their implementation is valid.

Cons:
* One concern I have is the lack of novelty in this paper. Whilst this is the first time (to my knowledge) a MuZero-style algorithm has been applied to the offline setting, most of the important algorithmic details have been covered in the past, such as MuZero itself, and the Reanalyze algorithm, which in and of itself is simply an extension of Dyna [3] (which is not cited) to their algorithmic approach..
* Another concern is the lack of generality in the derived insights. It has been known for some time that model-based methods with planning can improve sample efficiency and perform well on the offline domain [5,6], so the key takeaway is that this also applies to MCTS+function approx.+model-based methods. My main issue is that approaches like MuZero are definitely not the predominant approach due to their high computational requirements (144 TPU Hours per single Atari game!), and I am struggling to see what can be gleaned from this paper to more generic model based approaches.
* The baselines in the continuous control experiments seems lacking. For instance, there are other model-based methods that perform well on similar benchmarks (e.g., [5,6,7,8]) that are not compared with here; I believe the open source code is available for at least one of these. On a similar note, the results should be significance tested to demonstrate statistical power rather than just highlighting the largest number.

Comments/Nits:
* L65: “they do not directly use the learned model for planning over action sequences.”; I don’t think this is strictly true, since MBOP [5] does use the model for planning.
* L203: “value target based on a target network for the.”: unfinished sentence?

Conclusion:
Overall this paper shows the importance of trajectory reuse in model-based methods by applying it to MuZero (calling it Reanalyze), and shows that the resultant method provides excellent performance both fully online and fully offline, achieving state-of-the-art performance in an offline Atari benchmark. Furthermore, the scientific approach is generally good and the paper is fairly easy to follow.

However, I have concerns regarding the novelty of this work, as well as the applicability of the insights to a wider audience, as this work is so focussed on investigating the finer details of the MuZero algorithm. Furthermore, I believe more benchmarks should be provided for the continuous control experiments. Taken together, I therefore recommend a weak accept, but would like to see if the authors can provide more general insights, as well as more benchmarks for the DM Control section.

[1] Mastering Atari, Go, Chess and Shogi by Planning with a Learned Model, Schrittwieser et al.

[2] Value Prediction Network, Oh et al.

[3] Dyna, an integrated architecture for learning, planning, and reacting, Sutton

[4] Deep Reinforcement Learning in a Handful of Trials using Probabilistic Dynamics Models, Chua et al.

[5] Model-Based Offline Planning, Argenson and Dulac-Arnold

[6] MOReL: Model-Based Offline Reinforcement Learning, Kidambi et al.

[7] MOPO: Model-based Offline Policy Optimization, Yu et al.

[8] COMBO: Conservative Offline Model-Based Policy Optimization, Yu et al.

**Time Spent Reviewing:**

12

---

> ### Author Response · Authors · 2021-08-10
> **Online and Offline Reinforcement Learning by Planning with a Learned Model**
>
> Thank you for catching the missing citation of Dyna, we've corrected this oversight. However, Reanalyse is qualitatively different from Dyna in several important regards: it uses both value and policy rather than value function alone; and it also updates the state representation. In the specific case of MuZero Reanalyse it also performs a tree search rather than a single step lookahead used in Dyna.
>
> A major insight of our work is that a single unified approach can work well for both online and offline RL - something which starkly contrasts with previous work, which (as far as we are aware) has focused on treating these two settings separately. We hope that future work will continue this approach, tackle online and offline RL jointly and report results for both.
>
> Similarly, the result that using the MuZero model as an auxiliary loss improves the results of value based action selection (Table 2) should be broadly applicable to all value-based algorithms (even if they otherwise do not use planning or a model); for further discussion see Section 5, line 214 - 220.
>
> Further, the Reanalyse algorithm can be instantiated with any model-based improvement operator, not just MuZero / MCTS - not just to improve data efficiency, but also for other purposes, as described in Section 3 (page 4). For example, one could use MPO-like policy improvement operators such as those described in Muesli [1], and other planning methods such as single-step lookahead or model-predictive control [3]. It is also possible to use a sample-based improvement operator [2], such as the one used in the continuous action experiments.
>
> We would also like to point out that the MuZero algorithm itself does not have high computational requirements - as for other algorithms, the computational requirements depend on the choice of network architecture. Other algorithms that use the same network architecture (e.g  Muesli [1]) have similar computational requirements.
> Conversely, MuZero can use a smaller neural network for more efficient experiments, as in our experiments on the DM Control suite: 60 TPU hours per experiment (Section H), the majority of which was due to inefficiencies in our implementation and could be reduced further.
>
> We agree with the reviewer that comparing against even more other methods and on more domains would be useful. Unfortunately, we were constrained in the number of different domains and datasets we had time to integrate in our training setup. We therefore chose one dataset/benchmark per modality, settling on Atari and RL Unplugged, while MOReL, MOPO and COMBO only report results on D4RL.
> We will run multiple seeds for each experiment to compute confidence intervals and allow statistical testing as requested; we will update the paper with these results.
>
> [1] Matteo Hessel, Ivo Danihelka, Fabio Viola, Arthur Guez, Simon Schmitt, Laurent Sifre, Theophane Weber, David Silver, Hado Van Hasselt Proceedings of the 38th International Conference on Machine Learning, PMLR 139:4214-4226, 2021.
>
> [2] Thomas Hubert, Julian Schrittwieser, Ioannis Antonoglou, Mohammadamin Barekatain, Simon Schmitt, David Silver Proceedings of the 38th International Conference on Machine Learning, PMLR 139:4476-4486, 2021.
>
> [3] Kouvaritakis, Basil, and Mark Cannon. "Model predictive control." Switzerland: Springer International Publishing (2016): 38.

---

> > ### Comment · Reviewer_WqPe · 2021-08-27
> > **Thanks for the response.**
> >
> > Hi there,
> >
> > Point taken r.e. the qualitative differences w.r.t. Dyna. I think my point is more the spirit of re-using samples in this way is remarkably similar, but I now agree there are some differences, and these should be made clear in the paper.
> >
> > I agree with the authors about using MuZero as an auxiliary/self-supervised loss. Whilst I'm not convinced 60 TPU hours (even taking implementation inefficiencies into account) represents 'feasible' training for most people, I think my broader point about the generality of the findings has been sufficiently addressed.
> >
> > I am encouraged to hear that the authors will include additional results in the revision. I accept that these were not benchmarked on the RL Unplugged, and subsequently didn't penalize heavily for this.
> >
> > Taking this all into account, I believe this is now a valuable contribution to offline model-based RL (and have raised my score to reflect this), and presents an intriguing future direction whereby algorithms that holistically solve both the online and offline paradigm can exist, and maybe should be developed henceforth. Perhaps a nice follow on would be evaluating in the deployment-efficient setting [1].
> >
> > [1] Deployment-Efficient Reinforcement Learning via Model-Based Offline Optimization, Matsushima et al., 2020

---

### Official Review · Reviewer_zEEy · 2021-07-16

**Rating:** 6
**Confidence:** 3

**Summary:**

This work proposes the Reanalyse algorithm which uses model-based policy and value improvement operators to compute new improved training targets on existing data points, allowing efficient learning for data budgets varying by several orders of magnitude. The Reanalyse can also be used to learn entirely from demonstrations without any environment interactions, as in the
case of offline Reinforcement Learning (offline RL). The MuZero Unplugged algorithm provides a single unified algorithm for any data budget, including offline RL.

**Ethical Concerns:**

No.

**Limitations And Societal Impact:**

Yes.

**Main Review:**

The Reanalyse (or Muzero unplugged) algorithm shows strong empirical performances on the Atari games and 9 different tasks in the DM Control Suite. By repeatedly computing updated targets on previously collected data throughout training, the algorithm is able to largely improve the data efficiency comparing to previous methods. It can switch between online and offline settings without extra adjustment via choosing different reanalyze fractions.

Given all those advantages, I still have some questions. First, the Reanalyse algorithm seems very similar to the one that is described in the original MuZero paper (which diminishes the contribution). I can understand that the previous version can only improve data efficiency for discrete action space. Other than that, could you comment if there are any other differences? Also, for the look-ahead loss function $l_t(\theta)$ defined in (1), there is no $l_2$ regularity term comparing to the original MuZero paper. How could this possibly help?

Given the comprehensive empirical study provided, I would like to set my score to weak accept for the moment.

**Time Spent Reviewing:**

18 Hours.

---

> ### Author Response · Authors · 2021-08-10
> **Online and Offline Reinforcement Learning by Planning with a Learned Model**
>
> There are indeed several differences between the preliminary version of Reanalyse described in the MuZero paper, and the version in this paper:
> - generalization beyond data efficiency improvement to offline RL (section 5)
> - extension to very large discrete or continuous action spaces (section 6)
> - alternative uses of Reanalyse to learn from demonstrations and exploit good episodes (section 3)
> - pseudocode for implementing Reanalyse
> as well as a comprehensive set of results and ablations in different domains.
>
> The reason that there is no weight regularization term in the loss (1) is that for Adam with decoupled weight decay [1] this regularization is applied directly to the weights by the optimizer, not indirectly through the gradients (see also Section A of the supplementary materials and the pseudocode).
>
> [1] Loshchilov, I. and Hutter, F. Fixing weight decay regularization in adam. CoRR, abs/1711.05101, 2017. URL http://arxiv.org/abs/1711.05101.

---

> > ### Comment · Reviewer_zEEy · 2021-08-28
> > **Reply**
> >
> > Thank you for the reply! I am happy to keep the positive score for this paper.

---

### Official Review · Reviewer_7BJj · 2021-07-16

**Rating:** 8
**Confidence:** 4

**Summary:**

The authors explore in depth the reanalyze method specified in muzero[2] originally, in the context of data efficiency and offline RL. The reanalyze fraction can be changed to adapt for different dataset budgets. Both discrete and continuous variants of the muzero algorithm are explored for fully offline settings using RL Unplugged dataset/benchmarks.


**Ethics Review Area:**

["I don’t know"]

**Limitations And Societal Impact:**

Yes.

**Main Review:**

**Strengths:**

S1: Overall, the proposed method is straightforward, intuitive, and outperforms baselines on most of the challenging tasks.


S2: The experimental results consist of both online and offline settings across discrete and continuous action spaces.

S3: Addition of Limitations section and pseudocode in the supplementary materials both are very  welcome updates.

S4: The ablations study (Appendix table 9) on network sizes are welcome additions. It would be wonderful if a pointer to this was added in the description of table 3.

S5: The presentation quality is high. Figures are and tables clear/readable. Writing is clear.


**Overall** I am quite happy with the relative improvements of the paper and believe that the community will overall benefit from the provided insights and results.


**Minor Clarity Issues/ Comments:**

CL1: Line 52-55 may not hold true in light of new publications (While the overall notion of these kind of work being sparse is true, I feel it is worth mentioning other similar works). [1] [3]

** References:**

[1] Rafailov, Rafael et al. “Offline Reinforcement Learning from Images with Latent Space Models.” L4DC (2021).

[2] Schrittwieser, Julian et al. “Mastering Atari, Go, Chess and Shogi by Planning with a Learned Model.” Nature 588 7839 (2020): 604-609 .

[3] Shrestha, Aayam et al. “DeepAveragers: Offline Reinforcement Learning by Solving Derived Non-Parametric MDPs.” ArXiv abs/2010.08891 (2020): n. pag.

**Time Spent Reviewing:**

6

---

> ### Author Response · Authors · 2021-08-10
> **Online and Offline Reinforcement Learning by Planning with a Learned Model**
>
> Thank you for pointing out the paper from Shrestha, Aayam et al., we've added it to the discussion of previous work. We've also updated the citation of Rafailov et al. from the previous ArXiv submission to the more recent L4DC version.
>
> We referenced the model size ablation more directly in Table 3 as suggested.

---

### Author Response · Authors · 2021-08-10
**Thank you!**

We would like to thank all our reviewers for taking the time to read the paper in detail and writing such thorough and helpful reviews!

---

### Decision · Program_Chairs · 2021-09-27

**Decision:**

Accept (Spotlight)

**Comment:**

The authors have done a good job responding to the reviewers' questions. The reviewers are in consensus that the paper is a worthy contribution to the empirically aspects of offline reinforcement learning.

The authors are encouraged to include some discussion in relationship to the recent advances in the theory of offline RL, e.g., those from the recent workshops:

1. ICML RL Theory workshop:  https://lyang36.github.io/icml2021_rltheory/#papers
2. NeurIPS'20 Offline RL workshop:  https://offline-rl-neurips.github.io/

and the references therein. Some of the papers there might provide interesting theoretical insight into why model-based approaches are the way to go for offline RL.